# Molecular basis for the bifunctional Uba4–Urm1 sulfur-relay system in tRNA thiolation and ubiquitin-like conjugation

Marta Pabis[1,†] (iD), Martin Termathe[2,†] (iD), Keerthiraju E Ravichandran[1,3,†] (iD), Sandra D Kienast[2,4], Rościsław Krutyhołowa[1,5] (iD), Mikołaj Sokołowski[1,3] (iD), Urszula Jankowska[1] (iD), Przemysław Grudnik[1] (iD), Sebastian A Leidel[2,4,*] (iD) & Sebastian Glatt[1,**] (iD)

## Abstract

The chemical modification of tRNA bases by sulfur is crucial to tune translation and to optimize protein synthesis. In eukaryotes, the ubiquitin-related modifier 1 (Urm1) pathway is responsible for the synthesis of 2-thiolated wobble uridine ($U_{34}$). During the key step of the modification cascade, the E1-like activating enzyme ubiquitin-like protein activator 4 (Uba4) first adenylates and thiocarboxylates the C-terminus of its substrate Urm1. Subsequently, activated thiocarboxylated Urm1 (Urm1-COSH) can serve as a sulfur donor for specific tRNA thiolases or participate in ubiquitin-like conjugation reactions. Structural and mechanistic details of Uba4 and Urm1 have remained elusive but are key to understand the evolutionary branch point between ubiquitin-like proteins (UBL) and sulfur-relay systems. Here, we report the crystal structures of full-length Uba4 and its heterodimeric complex with its substrate Urm1. We show how the two domains of Uba4 orchestrate recognition, binding, and thiocarboxylation of the C-terminus of Urm1. Finally, we uncover how the catalytic domains of Uba4 communicate efficiently during the reaction cycle and identify a mechanism that enables Uba4 to protect itself against self-conjugation with its own product, namely activated Urm1-COSH.

**Keywords** adenylation; thioester; thiolation; tRNA modification; ubiquitin-like proteins

**Subject Categories** Post-translational Modifications & Proteolysis; RNA Biology; Structural Biology

**The EMBO Journal (2020) 39: e105087**

## Introduction

The Urm1 pathway is required for 2-thiolation of wobble uridines ($s^2U_{34}$) in eukaryotic tRNAs (Fig 1A). This modification is universally conserved and plays a pivotal role for translational fidelity, reading-frame maintenance, translation-elongation rates, and co-translation folding dynamics (Ranjan & Rodnina, 2016). $s^2U_{34}$ is always found along with 5-methoxycarbonylmethyl ($mcm^5$) modifications attached to the same base. In eukaryotes, this $mcm^5$ modification is introduced by the Elongator complex and Trm9 (Kalhor & Clarke, 2003; Huang et al, 2005; Chen et al, 2011; Dauden et al, 2019; Lin et al, 2019). Although $cm^5$-based $U_{34}$ modifications are found in 11 out of 13 tRNAs harboring wobble uridine, only $tRNA_{UUC}^{Glu}$, $tRNA_{UUG}^{Gln}$, and $tRNA_{UUU}^{Lys}$ in all eukaryotes and $tRNA_{UCU}^{Arg}$ in vertebrates carry $mcm^5s^2U_{34}$ (Huang et al, 2005; Yoshida et al, 2015; Schaffrath & Leidel, 2017). Dynamic changes in tRNA thiolation levels have been implicated in the cellular response to changing environmental conditions and in the regulation of sulfur, carbon, and nitrogen metabolic homeostasis (Laxman et al, 2013; Alings et al, 2015; Damon et al, 2015; Gupta et al, 2019). Finally, perturbations in $U_{34}$ modification levels trigger protein homeostasis defects and have been linked to the etiology of neurological disorders and cancer (Laguesse et al, 2015; Nedialkova & Leidel, 2015; Dauden et al, 2017; Rapino et al, 2017; Hawer et al, 2018).

Despite the fact that neither Uba4 nor Urm1 directly contact tRNA, their interaction and enzymatic activities are indispensable for $s^2U_{34}$ biosynthesis in all eukaryotes. Uba4 is a two-domain protein containing an adenylation domain (AD) and a rhodanese-like domain (RHD) (Fig 1B), which are both essential for the dual function of Uba4 in sulfur transfer and ubiquitin-like conjugation

1 Malopolska Centre of Biotechnology (MCB), Jagiellonian University, Krakow, Poland
2 Max Planck Institute for Molecular Biomedicine, Muenster, Germany
3 Postgraduate School of Molecular Medicine, Warsaw, Poland
4 Department of Chemistry and Biochemistry, University of Bern, Bern, Switzerland
5 Faculty of Biochemistry, Biophysics and Biotechnology, Jagiellonian University, Krakow, Poland
*Corresponding author. Tel: +41 031 631 42 96; E-mail: sebastian.leidel@dcb.unibe.ch
**Corresponding author. Tel: +48 012 664 6321; E-mail: sebastian.glatt@uj.edu.pl
†These authors contributed equally to this work

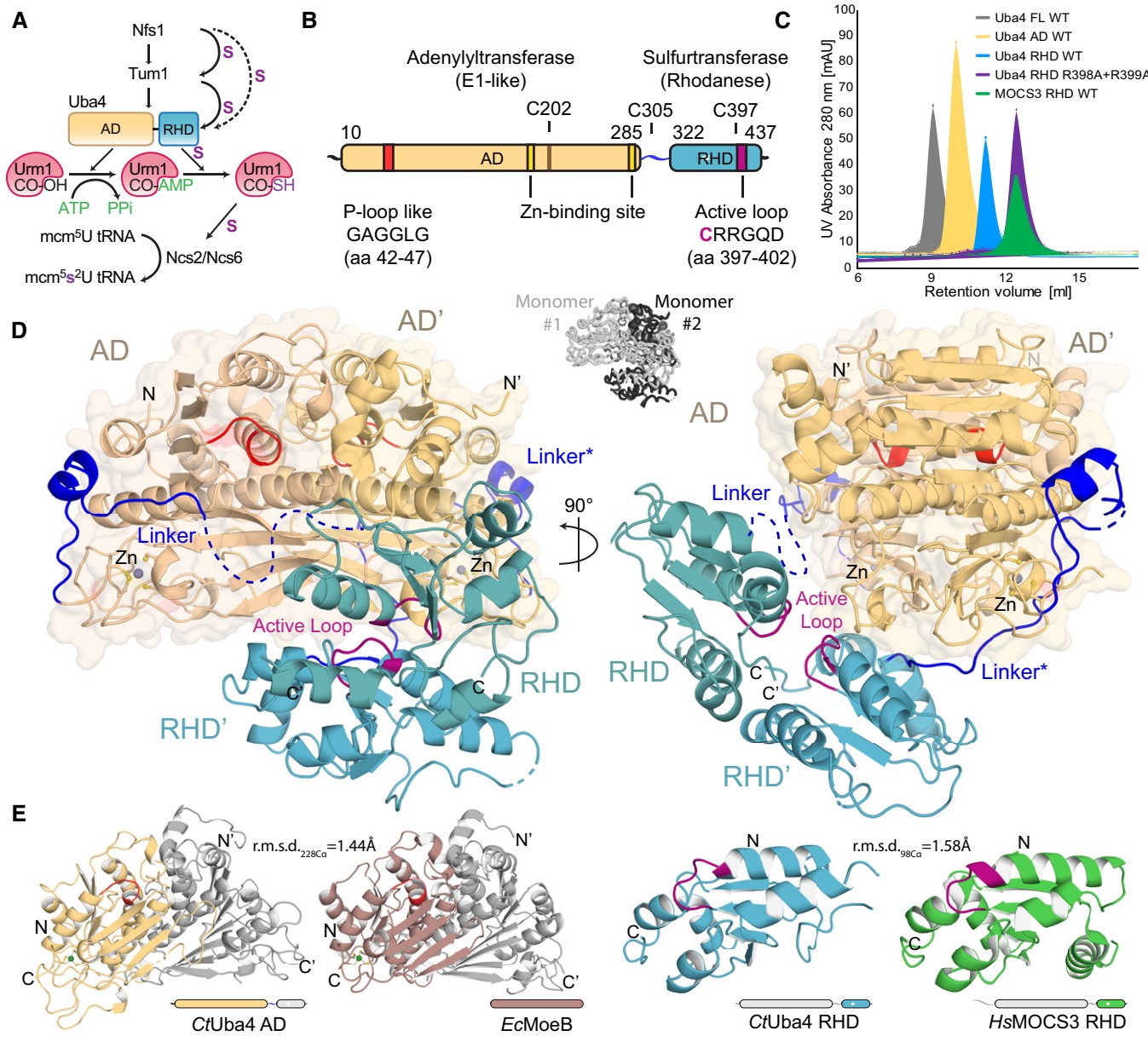

**Figure 1. Crystal structure of eukaryotic Uba4 at 2.2 Å resolution.**

A Scheme of the cytoplasmic tRNA thiolation pathway. Nfs1 mobilizes the sulfur and transfers it via a persulfide onto the rhodanese-like domains (RHDs) of either Tum1 or Uba4. Uba4 catalyzes the activation and thiocarboxylation of Urm1. Urm1-COSH is used by the Ncs2/Ncs6 complex to form 2-thiouridine on tRNAs. AD: adenylation domain. RHD: rhodanese-like domain.

B Domain organization of *Ct*Uba4. AD: adenylation domain. RHD: rhodanese-like domain.

C Size-exclusion chromatography profiles of recombinant proteins used in this study. AU: arbitrary units; WT: wild type; FL: full-length.

D Cartoon representation of the *Ct*Uba4 dimer. The AD dimer is emphasized by a transparent surface representation, and disordered loops are shown as dashed lines. The AD and RHD active loops are colored in red and purple, respectively. Zn atoms are shown as grey spheres. The two interwound Uba4 monomers are shown in black and white (central inset).

E Structural comparison of *Ct*Uba4 AD with *Ec*MoeB (PDB ID: 1JW9) and of *Ct*Uba4 RHD with *Hs*MOCS3 RHD (PDB ID: 3I2V). r.m.s.d. values as a measure of structural similarity are indicated.

(Furukawa *et al*, 2000; Huang *et al*, 2008; Schmitz *et al*, 2008; Leidel *et al*, 2009; Noma *et al*, 2009). Uba4 activates the C-terminus of Urm1 via an ATP-dependent adenylation step, receives mobilized sulfur from the Nfs1/Tum1-relay system by its RHD, and subsequently forms thiocarboxylated Urm1 (Urm1-COSH; Leidel *et al*,

2009; Noma *et al*, 2009). Thereafter, Urm1-COSH is released from Uba4 and acts as the sulfur donor for the Ncs2/Ncs6 heterodimer, which binds the tRNA targets to catalyze the actual 2-thiolation of $U_{34}$. Biochemical work has identified some features and reaction intermediates of the Uba4-mediated enzymatic reactions (Schmitz

*et al*, 2008; Termathe & Leidel, 2018), but the structure of Uba4 and in particular in complex with Urm1 that would enable a mechanistic understanding of their interplay is still missing. Foremost, how the intrinsically linked AD and RHD dynamically arrange to ensure their concerted enzymatic action during the reaction remains poorly understood.

Finally, Urm1 and Uba4 represent the most ancestral eukaryotic E1-UBL conjugation system that is currently known and most likely have served as a starting point for the emergence of all other UBL systems. Nonetheless, the biological function and physiological importance of protein "urmylation" and the identity of specific target proteins remain unclear (Furukawa *et al*, 2000; Goehring *et al*, 2003; Leidel *et al*, 2009; Van der Veen *et al*, 2011). Importantly, the molecular mechanism of Urm1 activation by Uba4 and the identity of its E2 and E3 enzymes have remained elusive, while structures are available for most other eukaryotic E1 enzymes. Therefore, the molecular details of the domain interaction, activation, and regulation of Uba4 are also critical to understand the diversification of well-known UBLs and sulfur carrier proteins (SCP) during early UBL evolution (Schmitz *et al*, 2008; Termathe & Leidel, 2018).

# Results

### Full-length Uba4 forms an asymmetric dimer

To understand how the Uba4–Urm1 system relates to SCP and UBL, we performed detailed structure–function analyses of its components. We heterologously expressed and purified Uba4 from *Chaetomium thermophilum* (*Ct*Uba4), a thermophilic fungus harboring highly stable proteins, well-suited for X-ray crystallography and other *in vitro* approaches (Bock *et al*, 2014; Fig 1C). Both, *Ct*Uba4 and its substrate *Ct*Urm1, display high sequence identity and similarity with their human orthologues (Appendix Fig S1). We solved the crystal structure of full-length *Ct*Uba4 at 2.2 Å resolution and refined its corresponding atomic model to $R/R_{free}$ values of 21.0%/25.3% (Table 1). Interestingly, the structure reveals a homo-dimeric complex of two protein units with an asymmetric positioning of the ADs and RHDs, which themselves form individual dimers (Fig 1D). While the ADs and RHDs individually form C2-symmetrical dimers, the structure is asymmetric since the two 2-fold axes do not coincide. To independently confirm the unexpected dimeric form of Uba4, we expressed the AD and the RHD separately and analyzed the purified proteins using size-exclusion chromatography (SEC) and static light scattering in solution (Figs 1C and EV1A). Indeed, both domains form closely interacting dimers on their own. While members of the RHD family occur in variable arrangements, Uba4 is unique as it is the only protein, where a RHD is physically connected to an AD by a short linker of unknown function. The individual ADs are formed by a continuous sheet of eight β-strands surrounded by eight α-helices and are highly similar to both MoeB, the prokaryotic ancestor of Uba4, and to ADs of UBL E1 enzymes (Cappadocia & Lima, 2018; Lake *et al*, 2001; Olsen *et al*, 2010; Fig 1E). The exposed AD surface is predominantly positively charged, while the dimer interface is mostly hydrophobic and shows a high degree of conservation (Appendix Fig S2). Like the AD, the RHD forms a dimer mediated by hydrogen bonds between active-

site loop residues Arg398, Arg399, and Glu405 occluding the active cysteines (Fig EV1B), which suggests that the RHDs are inhibited in the apo form of Uba4. To test the importance of Arg398 and Arg399 for dimerization, we mutated both residues and found that RHD$_{R398A/R399A}$ behaves as a monomer in solution (Figs 1C and EV1A). The Uba4 RHD dimer shows a relatively small interface of 706.8 Å$^2$ and a unique arrangement compared to other known RHD dimers within the context of the full Uba4 dimer (Fig EV1C). Noteworthy, the RHD dimer mainly interacts with only one AD (Fig 1D). In summary, our data show that full-length Uba4 forms an asymmetric homodimer that the AD and the RHD independently contribute to this unique apo assembly that likely inactivates the RHD.

### Full-length Uba4 adenylates Urm1

The asymmetry of the Uba4 dimer has important functional implications. One Uba4 dimer harbors two nucleotide-binding sites, formed

**Table 1.  Data collection and refinement statistics (molecular replacement).**

|  | *Ct*Uba4 PDB ID 6YUB | *Ct*Uba4-*Ct*Urm1 PDB ID 6YUC |
|---|---|---|
| **Data collection** | P11 DESY | XRD2 Elettra |
| Wavelength (Å) | 1.0332 | 1.0 |
| Space group | P121 (4) | R32 (155) |
| Cell dimensions *a, b, c* (Å) | 64.48 73.96 103.36 | 197.74 197.74 99.57 |
| α, β, γ (°) | 90.0 106.1 90.0 | 90.0 90.0 120.0 |
| Resolution (Å) | 50–2.195 (2.274–2.195)[a] | 49.43–3.15 (3.34–3.15) |
| $R_{meas}$ (%) | 14.12 (135.6) | 6.6 (472.2) |
| $R_{pim}$ (%) | 5.4 (53.54) | 2.1 (191.4) |
| $I/\sigma I$ | 10.33 (1.31) | 18.85 (0.48) |
| CC1/2 | 0.997 (0.689) | 1 (0.297) |
| Completeness (%) | 99.44 (95.59) | 99.8 (99.2) |
| Redundancy | 6.6 (6.2) | 9.86 (9.43) |
| **Refinement** |  |  |
| Resolution (Å) | 47.623–2.195 | 42.867–3.153 |
| No. reflections | 48,062 | 9,580 |
| $R_{work}/R_{free}$ | 0.2103/0.2526 | 0.2103/0.2631 |
| No. atoms | 6,097 | 2,878 |
| Protein | 6,097 | 2,877 |
| Ligand/ion | 2 | 1 |
| Water | 224 | 0 |
| *B*-factors (Å$^2$) | 53.98 | 156.74 |
| Protein | 54.04 | 156.76 |
| Ligand/ion | 41.72 | 110.05 |
| Water | 52.45 | N.A. |
| R.m.s. deviations |  |  |
| Bond lengths (Å) | 0.006 | 0.006 |
| Bond angles (°) | 1.03 | 1.52 |

[a]Values in parentheses are for highest-resolution shell.

by conserved residues of both ADs (Gly45, Arg77, Lys90, Asp134 of one subunit and Arg18 of the second subunit) implying that AD homodimerization is essential for the adenylation of Urm1 (Fig EV2A). This arrangement of the ADs distinguishes Uba4 from canonical E1 enzymes (e.g., Uba1, Uba2, Uba3) that harbor only one active adenylation domain and resembles the prokaryotic MoeB (Lake *et al*, 2001) and the eukaryotic Atg7 (Schulman & Harper, 2009). Although Uba4 was crystallized in the absence of nucleotides,

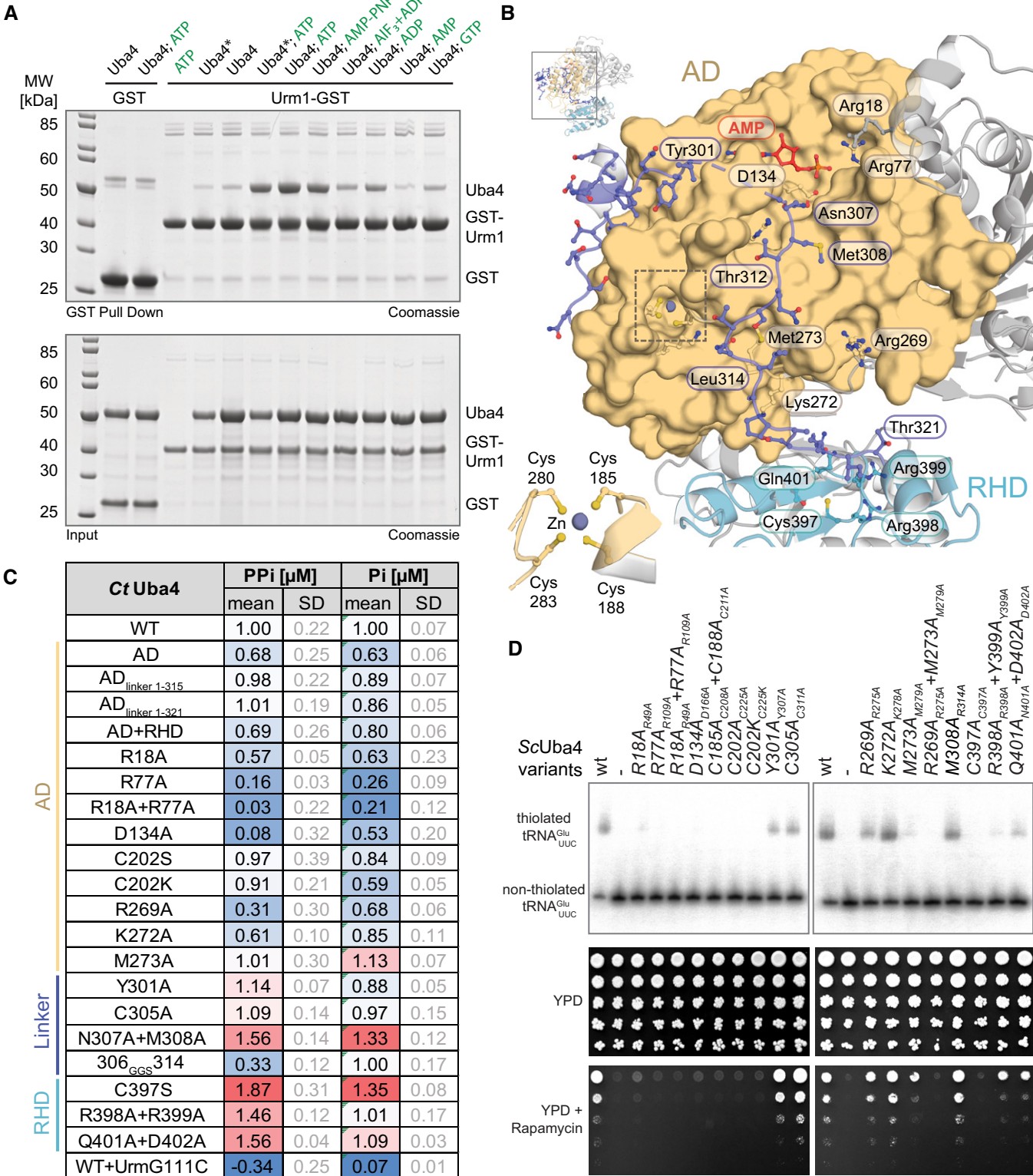

**Figure 2.**

◀

Figure 2.  **Functional analysis of structural elements in Uba4.**

A   Analysis of the interaction between *Ct*Uba4 and GST-*Ct*Urm1 (ratio 2:1 or 0.5:1 when marked with an asterisk) in the presence of nucleotide derivates by GST pull-down resolved by SDS–PAGE and visualized with Coomassie stain.

B   Close-up (for reference see full structure on upper left) of the *Ct*Uba4 linker (aa 286–321) highlighting important residues and the coordination of the zinc site (inset, lower left).

C   Tabular summary of ATP hydrolysis by *Ct*Uba4 WT and mutants in the presence of *Ct*Urm1. The amount of released PPi was measured using a fluorometric pyrophosphate assay kit. After PPi hydrolysis, Pi was also detected by a Malachite Green kit. Changes in enzymatic activity are indicated as a color gradient from blue to red, normalized to *Ct*Uba4 WT. PPi: pyrophosphate; Pi: inorganic phosphate; SD: standard deviation; $n \geq 3$.

D   Functional analyses of *Sc*Uba4 mutant yeast using APM-gel retardation (top) and viability in response to rapamycin (bottom). PAGE was supplemented with APM to retard the migration of thiolated tRNAs and allow their visualization by Northern blot using an anti-tRNA$_{UUC}^{Glu}$ probe. APM: ([*N*-Acryloyl-amino]phenyl)mercuric chloride; YPD: yeast extract peptone dextrose. All residue numbering follows the *Ct*Uba4 sequence, but the respective *Sc*Uba4 numbering is added in subscript.

the highly conserved binding pocket enabled us to model an AMP molecule into the structure with high confidence. To understand how nucleotide binding modulates Uba4 activity, we analyzed the effect of a panel of nucleotides on the recruitment of Urm1 by Uba4 (Fig 2A). We found that the Uba4–Urm1 complex is stabilized most efficiently in the presence of ATP and AMP-PNP, an ATP analogue with a non-hydrolyzable β-γ-imido bond. Other nucleotides and nucleotide derivatives (e.g., AMP and GTP) led to very weak or completely diminished complex formation (Fig 2A). AMP increased the thermal stability of full-length Uba4 and the AD alone, indicating that an AMP-related intermediate, like the adenylated form of the Urm1 C-terminus induces a more compact conformation of the AD (Appendix Table S1). Urm1 lacking the C-terminal glycine residue (Urm1$_{G111C}$) did not trigger ATP hydrolysis (Appendix Table S2). Thus, initial ATP binding (Haas & Rose, 1982) is followed by Urm1 binding, ATP hydrolysis, and subsequent adenylation. Stable Uba4–Urm1 complex formation requires the combined presence of Uba4, Urm1, and ATP in the reaction, which is in contrast to other known canonical and non-canonical E1 enzymes that are able to bind their UBL also in the absence of ATP and $Mg^{2+}$ ions (Lake *et al*, 2001; Noda *et al*, 2011; Oweis *et al*, 2016).

A defining feature of eukaryotic Uba4 proteins is that their E1-like AD and RHD are covalently connected by a flexible linker region. However, the function of this linker is undefined. We found that in one of the two Uba4 molecules, almost the entire linker adopts a stable conformation with a well-defined electron density (Figs 2B and EV2B), while this is not the case for the second Uba4 molecule in the dimer. The structured linker closely interacts with the AD, and its residues, in particular Asn307 and Met308, are in close proximity to the ATP-binding pocket, coinciding with the expected localization of the C-terminus of Urm1 when bound to Uba4 (Fig 2B). In contrast, only the first residues of the second linker are stably associated with the AD via the Zn-binding site and a strong hydrogen-bond network, while the majority of this linker appears flexible and dynamic thereby permitting the asymmetric positioning of the RHD with respect to the AD dimer (Fig 1D). Thus, the two linkers appear to ensure (i) a regulated binding of ATP and Urm1 by restricting the binding area and (ii) an optimal asymmetric positioning of the RHD in the absence of Urm1 binding.

## Uba4 mutations affect U$_{34}$ thiolation and define a functional role for the linker region

To test the functional importance of specific residues of the AD, the RHD, and the linker (Fig EV2C) that we identified in our structure, we combined *in vitro* *Ct*Uba4 activity assays with functional

*in vivo* validation in baker's yeast (Appendix Table S3). This allowed us to correlate the enzymatic activity of Uba4 (Fig 2C), intracellular tRNA thiolation levels, and stress sensitivity in order to determine *in vivo* functionality (Fig 2D and Appendix Fig S3). As expected, mutations in the AD nucleotide-binding site (R18A, R77A and D134A) strongly reduced the adenylyltransferase activity *in vitro* (Fig 2C and Appendix Table S2), abrogated tRNA thiolation in yeast (R49A, R109A, and D166A), and induced rapamycin sensitivity in these strains (Fig 2D). Similarly, disruption of the structural zinc-binding site (C208A/C211A) had a strong functional effect *in vivo*, likely due to decreased stability of the protein resulting in lower levels of available protein and reduced adenylation efficiency (Appendix Fig S3B; Wang & Chen, 2010). Mutations of the catalytic cysteine Cys202, as well as conserved linker residues that are distant from the ATP-binding pocket (C305A, Y301A, M273A), do not affect ATP hydrolysis *in vitro*, establishing that these residues are not required for priming the adenylation reaction. However, mutating residues that are required for positioning of the linker and presumably the Urm1 C-terminus (e.g., R269A, K272A) decreased adenylyltransferase activity, confirming the requirement for Urm1 recruitment (Fig 2C). The corresponding combination of R275A and M279A led to a complete loss of thiolation accompanied with no growth in the presence of rapamycin in baker's yeast, while mutations of the linker residues Asn307 and Met308 located adjacent to the nucleotide-binding site led to an increase in ATP hydrolysis. This suggests that first, these residues act as a gate that is modulated by Urm1 binding and second, that the linker facilitates positioning of Asp134 for optimal control of the adenylation activity. The N307A/M308A mutation may therefore mimic Urm1 binding and augment the accessibility of the active site for ATP hydrolysis. The fact that tRNAs in the R314A mutant are modified and that the mutant grows like wild type supports this idea. Uba4$_{306GGS314}$, where we replaced nine linker residues by three Gly-Gly-Ser repeats, consumes ATP like the wild-type protein, but directly produces inorganic phosphate without generating the pyrophosphate (PPi) intermediate (Fig 2C). Furthermore, we attached the linker residues that are expected to compete with Urm1 binding (AD$_{linker\ 1–315}$) and the complete linker region (AD$_{linker\ 1–321}$) to the AD and compared their stability and their activity to full-length Uba4 and the AD alone. We show that the linker region strongly enhances the thermostability of the AD domain in the absence of Urm1 and that it stimulates the ATP hydrolysis activity of the AD in the presence of Urm1 (Fig 2C and Appendix Table S1). Our data show that the linker region is critical for the catalytic activity of Uba4, which loses its ability to hydrolyze ATP directly to AMP and PPi when the linker structure is perturbed. Finally, mutations of the RHD active loop, and in particular of

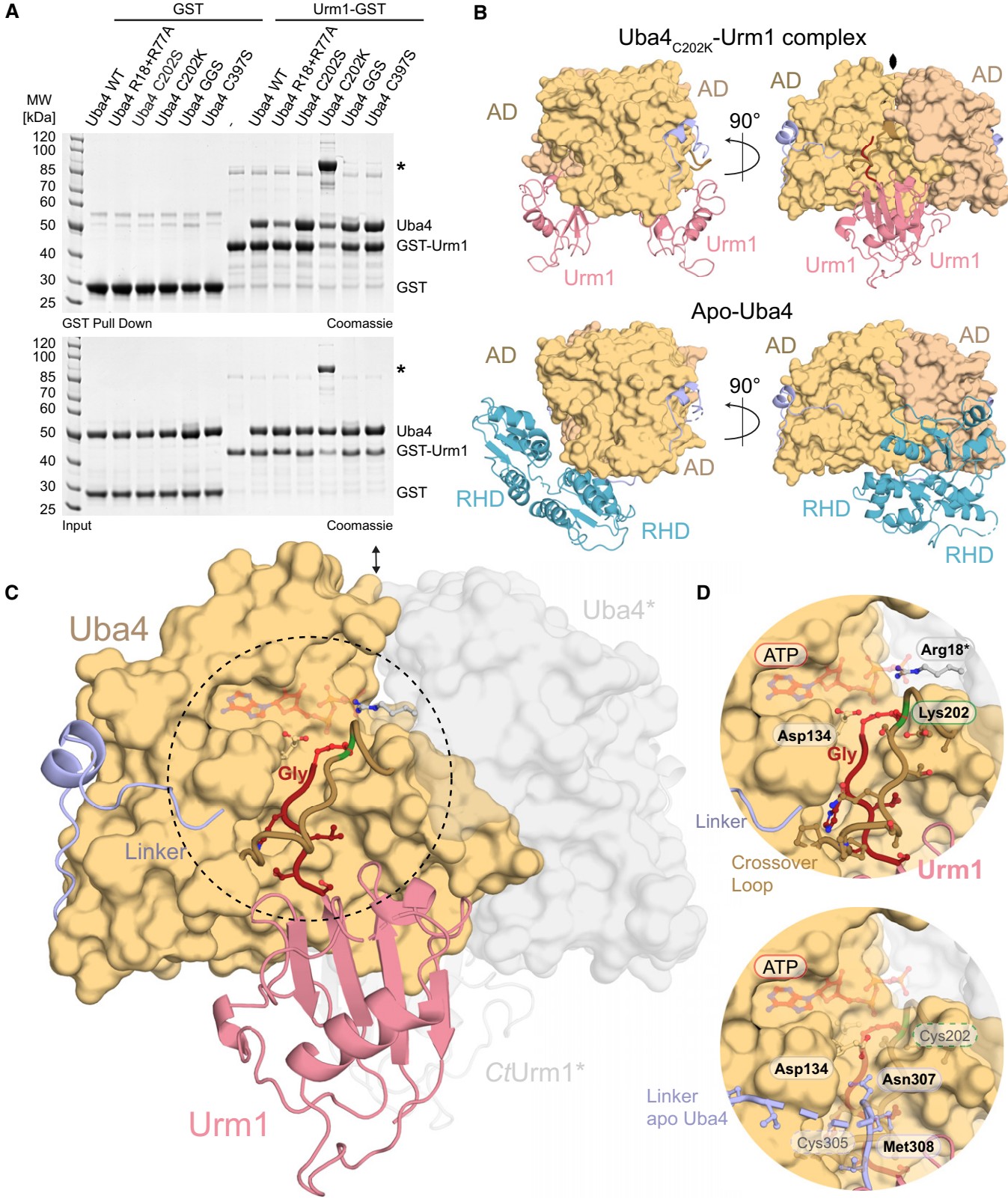

**Figure 3.**

**Figure 3. Crystal structure of the Uba4$_{C202K}$–Urm1 complex at 3.1 Å resolution.**

A   Interaction analysis of WT and mutated *Ct*Uba4 with GST-*Ct*Urm1 by GST pull-down in the presence of 1 mM ATP. *Ct*Uba4$_{C202K}$ covalently linked to GST-*Ct*Urm1 is marked by an asterisk.

B   Comparison of the structures of the *Ct*Uba4$_{C202K}$-*Ct*Urm1 complex and apo *Ct*Uba4 focusing on the arrangement of Urm1 and RHD in relation to the AD dimers. The symmetry operator of the crystallographic twofold axis is indicated.

C   Overall structure of the *Ct*Uba4$_{C202K}$-*Ct*Urm1 complex. The AD dimer is shown as surface representation, and the Uba4 linker (light blue), crossover loop (light brown), and Urm1 (salmon) are shown as cartoon. The C-terminus of Urm1 (dark red), Lys202 (green), and the formed covalent peptide bond between the two components (red) are highlighted. The ATP molecule (transparent red) is positioned in the nucleotide-binding site based on the ATP-bound *Ec*MoeB-MoaD complex (PDB ID: 1JWA). Uba4 and Urm1 molecules from the adjacent asymmetric unit which form the dimer are shown in transparent grey.

D   Close-up view of the *Ct*Uba4 active site and the *Ct*Urm1 C-terminus (top). A superposition of the linker from apo *Ct*Uba4 structure is shown (bottom). Crucial residues are shown as ball-and-stick model, and the tentative positions of Cys202 Cys305 are indicated.

catalytic Cys397, stimulate ATP hydrolysis by Uba4 (Fig 2C) and Uba4–Urm1 complex formation (Appendix Fig S4 and Appendix Table S2), suggesting a close interplay between the RHD active site and the Urm1 activation site in the AD. Taken together, our results establish that the linker is more than a simple connector between AD and RHD. The linker is a crucial part of the Uba4 reaction cycle and an integrated switch to coordinate nucleotide hydrolysis, substrate binding, Urm1 adenylation, and thiocarboxylation.

Finally, we used the set of Uba4 mutants to identify the specific reaction intermediate that stabilizes the Uba4–Urm1 complex after addition of ATP and AMP-PNP (Fig 2A). The previous observation of a hydrolysis event between the α- and β-phosphate and the production of pyrophosphate corroborates the role of Urm1 adenylation for the complex formation. Mutants lacking ATP hydrolysis activity (R18A+R77A and D134A+Q164A) indeed show strongly decreased ATP-dependent Urm1 binding, whereas mutations of the active-site cysteine that is known to form a thioester (C202S, C202A, and C202K) are not affected. Therefore, we conclude that the adenylation of Urm1 is sufficient to stabilize the complex between Uba4 and Urm1 in the presence of ATP (Figs EV2D and 3A, and Appendix Fig S4).

**Structure of the Uba4$_{C202K}$–Urm1 complex**

While the analysis of the Uba4 structure provided important insights into Urm1 adenylation, it did not reveal how the Uba4–Urm1 complex forms during activation. Therefore, we sought to determine the structure of the transition state of Urm1 activation by focusing on the complex formed between adenylated Urm1 (Urm1-AMP) bound in the active site of the AD. Since we have observed an increased efficiency and stability of the adenylated complex formed with RHD active loop mutants, we performed crystallization trials with Uba4, Uba4$_{C397S}$, Uba4$_{C202S/C397S}$, Uba4$_{R398A/R399A}$, and Uba4$_{Q401A/D402A}$ in complex with Urm1. However, all tested crystals exhibited only weak diffraction properties. We have recently shown that the formation of a thioester linkage between Uba4 and activated Urm1 is a key intermediate step prior to thiocarboxylation in baker's yeast (Termathe & Leidel, 2018). Therefore, we generated a Uba4$_{C202K}$ mutation to convert the labile thioester intermediate into a non-hydrolysable iso-peptide bond between the activated C-terminus of Urm1 and the ε-amino group of the introduced lysine. A similar strategy has been used successfully to trap stable E2-Ubiquitin complexes for structural studies (Plechanovová *et al*, 2012), but has never been applied to an E1 enzyme and its UBL. Indeed, we found that Uba4$_{C202K}$ forms a covalent bond with Urm1 in the presence of ATP, while other Uba4 mutants (C202S, C397S,

306$_{GGS}$314) led to the formation of non-covalent Uba4–Urm1 complexes similar to wild-type Uba4 (Fig 3A).

The covalently linked Uba4$_{C202K}$–Urm1 complex formed well-diffracting crystals resulting in a structure at 3.15 Å resolution, and its atomic model was refined to R/R$_{free}$ values of 21.1%/26.2% (Table 1). The structure revealed two Urm1 molecules with slightly higher B-factors bound to a dimer of the Uba4 AD (Figs 3B and EV3A), while no specific density could be attributed to the RHD, which although present in our crystals, appears to be mobile at this specific reaction step. This stoichiometry of the Uba4–Urm1 complex is in full agreement with the one we determined in solution by static light scattering (RALS/LALS) (Fig EV1A).

In contrast to Uba4 alone, the Uba4–Urm1 dimer is fully symmetric and the AD interface falls on a twofold crystallographic axis (Fig 3C). Interestingly, the C-terminal GG-motif of Urm1 is positioned in the same groove that was occupied by the linker region in the unbound Uba4 structure (Figs 3D and EV3B). This suggests that upon binding, Urm1 displaces the C-terminal linker region from the one Uba4 molecule where it is bound and promotes the flexibility of the RHD dimer. The symmetric nature of the Uba4–Urm1 complex and the mobility of the linker and the RHD form the basis of a hypothetical model where the RHD dimer that is observed in the Uba4 apo structure is disrupted upon Urm1 binding (Fig 3C). Subsequently, the individual monomeric RHDs may move toward the closest Urm1-bound AD. Furthermore, Urm1-induced monomerization of the RHD could simultaneously trigger exposure of the active-site cysteine (Cys397), which is buried in the RHD dimer interface and inaccessible in the Uba4 apo structure. Importantly, a helical region localized adjacent to the ATP-binding site (aa 72–88) in the structure of apo Uba4 becomes disordered and flexible in the structure of the Uba4–Urm1 complex (Fig EV4). This is reminiscent of the dissolution of the H1 and H2 helices in the structure of SUMO E1-SUMO-AVSN (Olsen *et al*, 2010), confirming that our strategy indeed locked the anticipated post-adenylation thioester state following nucleotide release. In contrast, the loop containing Cys202, known as "crossover loop" in other E1 enzymes, becomes structured in the Uba4–Urm1 complex and an electron density can be clearly assigned to the covalent peptide bond between the introduced Lys202 and the C-terminal glycine (Gly111) of Urm1 (Fig EV3B). In the Uba4 apo structure, the respective amino acids (aa 194–205) show no defined electron density due to intrinsic flexibility. Hence, the crossover loop acts as a clamp toward the C-terminus of Urm1 (Fig 3D), which appears to remain virtually identically bound to the AD throughout all reaction steps, including adenylation, thioester formation, and possibly thiocarboxylation.

## Uba4 protects itself against Urm1-COSH

Since Urm1 is located at the branchpoint of UBL and SCP evolution, it simultaneously serves two purposes. In addition to its dominant physiological role as the SCP in tRNA 2-thiolation, Urm1 was found to be covalently attached to target proteins via lysine residues upon

oxidative stress, similar to ubiquitin and other UBLs (Goehring *et al*, 2003; Van der Veen *et al*, 2011). However, Uba4 lacks ubiquitin-fold domains (UFD) and no specific conjugating E2 enzymes or E3 ligases have been described for Urm1. Therefore, it remains unresolved whether downstream conjugating enzymes are recruited to Uba4 at all (Kerscher *et al*, 2006). Nonetheless, MOCS3, the human

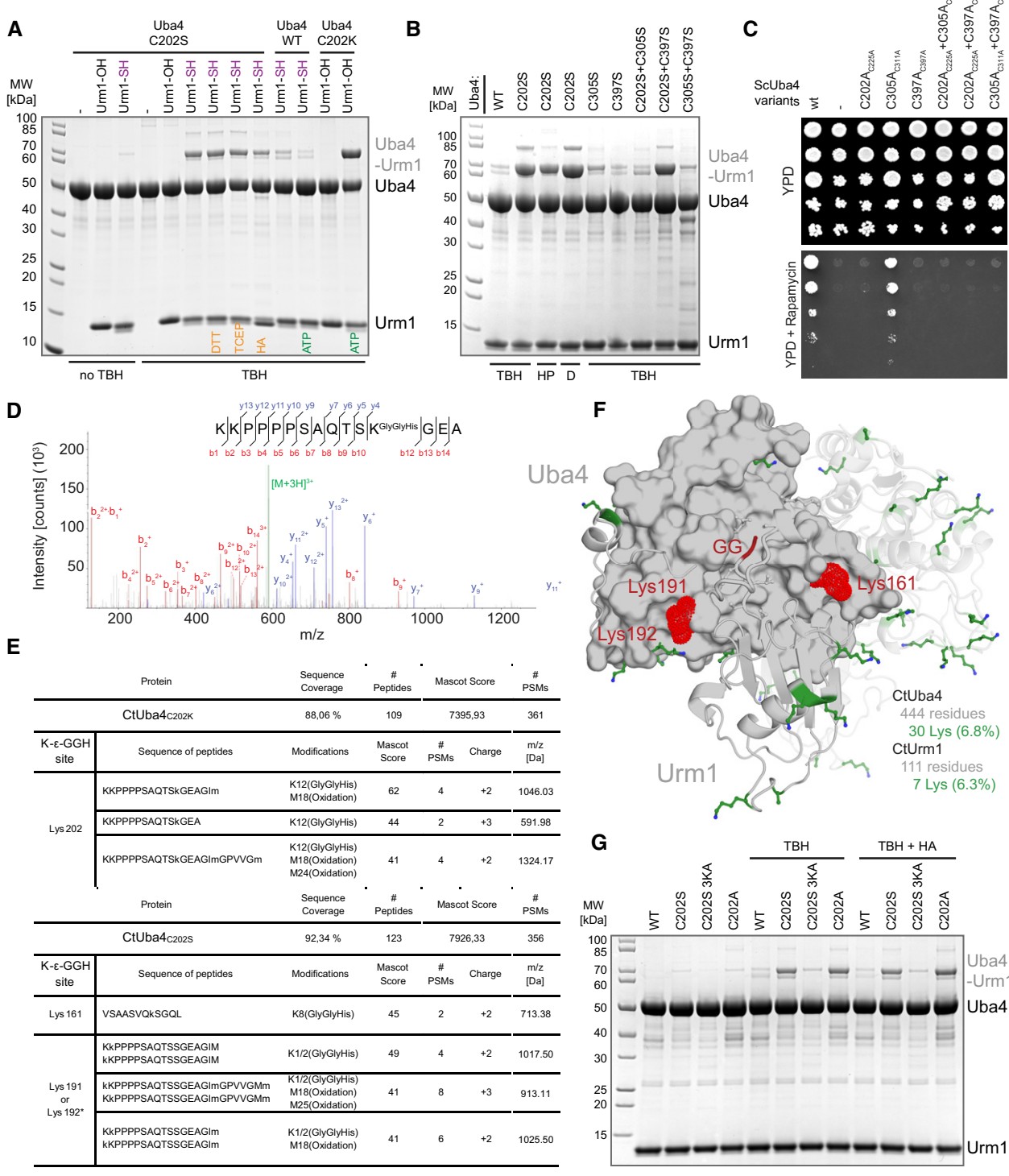

**Figure 4.**

**Figure 4. The active cysteine of the Uba4 AD protects against covalent attachment of thiocarboxylated Urm1.**

A   Analysis of covalent adduct formation between *Ct*Uba4 WT or C202 mutants and carboxylated (–OH) or thiocarboxylated (–SH) *Ct*Urm1 in the presence or absence of TBH, DTT, TCEP, and HA, respectively. ATP: Adenosine triphosphate TBH: *tert*-Butyl hydroperoxide HA: Hydroxylamine; DTT: 1,4-Dithiothreitol; TCEP: Tris(2-carboxyethyl)phosphine.

B   Analysis of covalent adduct formation between *Ct*Uba4 WT/cysteine mutants and *Ct*Urm1-COSH in the presence of oxidizing agents. HP: Hydrogen peroxide; D: Diamide.

C   Phenotypic analysis of *Sc*Uba4 mutant yeast strains in response to rapamycin.

D   MS/MS spectrum of the K202-ε-GGH containing peptide from *Ct*Uba4$_{C202K}$. A representative annotated fragmentation spectrum is shown with the b- and y-ions marked in red and blue, respectively. The precursor ion at $m/z$ 591,98 is labeled in green. The peptide sequence is shown at the top with the collision-induced fragmentation pattern. The Urm1 conjugation site was identified by detection of the GGH remnant motif generated by chymotryptic digestion.

E   The table shows parameters of mass spectrometry identification of *Ct*Uba4$_{C202K}$ and *Ct*Uba4$_{C202S}$ with lists of detected peptides containing the K-ε-GGH remnant motif of Urm1. In *Ct*Uba4$_{C202K}$, the conjugation site on K202 was confirmed. In the TBH treated samples of *Ct*Uba4$_{C202S}$, several potential conjugation sites were found. We present only high confidence peptides reaching a Mascot score higher than 40, which were detected in all analyzed samples. Asterisks indicate ambiguous site localization of K-ε-GGH. #PSMs: number of peptide-to-spectrum matches; $m/z$: mass-to-charge ratio of precursor ion.

F   Localization of all lysine residues (shown as green sticks) on the structure of *Ct*Uba4$_{C202K}$-Urm1. *Ct*Urm1 C-terminus and the lysine residues of *Ct*Uba4$_{C202S}$ that become covalently linked to Urm1-COSH in oxidizing conditions are presented in red.

G   Analysis of covalent adduct formation between *Ct*Uba4 WT or Cys202 mutants and thiocarboxylated (–SH) CtUrm1 in the presence or absence of TBH and HA, respectively.

Uba4 homologue, was detected as an Urm1 target in human cells (Van der Veen *et al*, 2011; Judes *et al*, 2015). Hence, we tested whether Urm1 can be covalently conjugated to Uba4 *in vitro*. Essentially, no Urm1 conjugation is observed when wild-type Uba4 is incubated with Urm1-OH or Urm1-COSH in the absence of oxidizing agents (Figs 4A and EV5A). Nonetheless, detectable levels of Uba4 are conjugated to Urm1-COSH (but not Urm1-COOH) when the reaction is supplemented with the oxidizing agent *tert*-Butyl hydroperoxide (TBH) (Fig 4A). The appearance of these conjugates is strongly enhanced by using Uba4$_{C202S}$. The highly abundant conjugates show the same molecular weight as the locked Uba4$_{C202K}$-Urm1 complex indicating an equimolar complex. Next, we treated the conjugates with (i) 1,4-dithiothreitol (DTT; resolves disulfide and thioester bonds), (ii) Tris(2-carboxyethyl)phosphine (TCEP; resolves disulfide bonds), or (iii) hydroxylamine (HA; resolves thioester bonds) (Termathe & Leidel, 2018). None of these reagents disrupted the conjugates, showing that a covalent iso-peptide bond is formed between Urm1-COSH and Uba4$_{C202S}$. The use of other oxidizing agents like hydrogen peroxide and diamide induces a similar conjugation reaction (Fig 4B). Surprisingly, Urm1 conjugation is reduced to wild-type levels in the Uba4$_{C202S/C305S}$ double mutant (Fig 4B), indicating a crucial redox communication between these two cysteines both located in adjacent loop regions (Fig 3D). Therefore, we tested single and double cysteine mutants in yeast for thiolation and growth defects (Figs 4C and EV5B). The experiments revealed that Cys311 is not essential for Uba4 function *in vivo,* although tRNA thiolation levels are slightly reduced (Figs 2D and 4C). Furthermore, we confirmed that neither C305K nor C397K lead to a locked intermediate after adenylation of Urm1 (Fig EV5C).

Finally, we used mass spectrometry to identify the lysine residue in Uba4, that is the conjugation site of Urm1 in the C202S mutant. We used Uba4$_{C202K}$ as a positive control showing exclusively peptides that link the Urm1 C-terminus and Lys202 of Uba4$_{C202K}$ (Fig 4D and Appendix Table S4). Interestingly, we identified a set of covalently linked lysine residues (Lys161, Lys191, and Lys192) in the TBH-treated samples of Uba4$_{C202S}$ (Fig 4E) close to the active site (Fig 4F). Furthermore, we show that a variant of C202S mutant lacking the lysine residues C202S/K161A/K191A/K192A shows a strong reduction in the Urm1 conjugates (Fig 4G). Our data confirm that Urm1-COSH is covalently attached in the absence of Cys202

with a low selectivity of the C202S-triggered conjugation reaction. Whereas Uba4$_{C202S}$ and an equivalent Uba4$_{C202A}$ mutant show enhanced conjugation (Fig 4G), substitutions of other active site or surface cysteines (i.e., C305S, C397S) do not promote the formation of Uba4–Urm1 conjugates (Figs 4B, and EV5D and E). This finding highlights the central role of Cys202, in resolving or reverting reactive intermediates during oxidative stress. These findings emphasize the importance of Cys202 and its precise positioning for the activation of the Urm1 C-terminus via a thioester and for the efficient release of reactive sulfur intermediates by Cys305. In addition, the self-conjugation of Uba4 by Urm1 represents the first evidence *in vitro* for the existence of an ubiquitin-like "urmylation" reaction.

## Discussion

2-Thiolation of U$_{34}$ is a universal tRNA modification that is found in all species and critical in the context of neurodegeneration and cancer (Grosjean *et al*, 2010; Torres *et al*, 2014; Grosjean & Westhof, 2016; Schaffrath & Leidel, 2017). However, the enzymatic machineries that place these modifications differ between bacteria and eukaryotes and many of the mechanistic steps of these processes have remained unclear. In particular, the lack of structural information of its key players has prevented us from considering the Urm1 pathway as a target for small molecule inhibitors that could be used to treat, e.g., BRAF therapy-resistant melanoma cancer cells (Rapino *et al*, 2018). By determining the first structures of the Uba4–Urm1 activation cycle, we closed this gap and discovered an intricate interplay of the AD and the RHD, which is guided by the hitherto uncharacterized linker that connects the two domains. Our data allow us to draw a mechanistic reaction scheme of the individual steps leading to Urm1 thiocarboxylation by Uba4 and to propose the structural rearrangements in the AD and RHD domains upon substrate binding and product release (Fig 5A).

In detail, we find that the AD and the RHD act independently of each other. However, the two domains of the eukaryotic Uba4 protein are genetically fused to each other, which requires that their enzymatic activities are intrinsically coordinated. The binding of the Urm1 substrate to the AD enforces the displacement of the bound linker region, which could lead to the separation of the RHD dimer

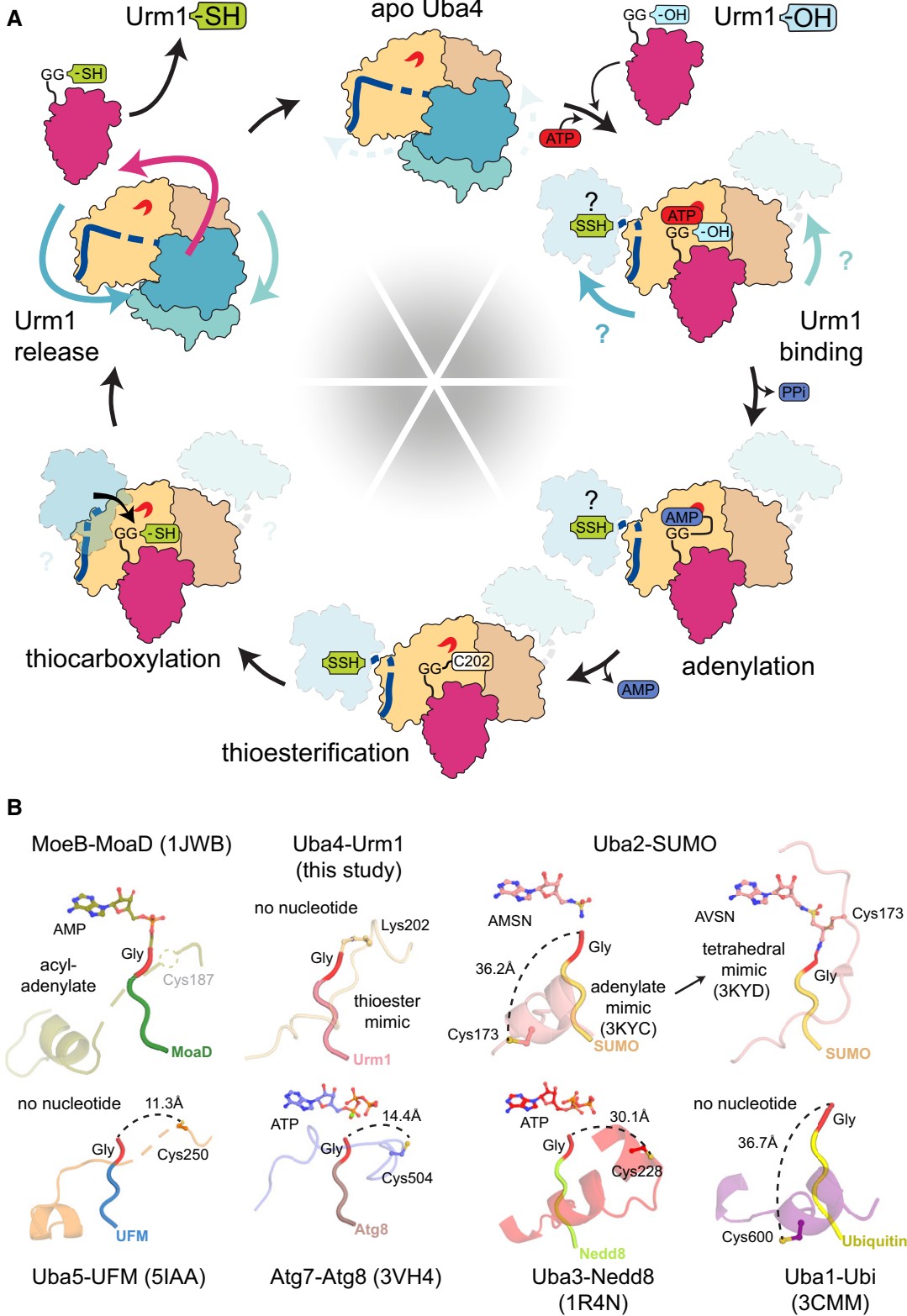

Figure 5.

and structural rearrangements of the RHD domains. These relative domain movements toggle the Uba4 dimer between an unloaded asymmetric and a substrate-bound symmetric form. As ATP binding

and hydrolysis activity of the AD are required to stably recruit Urm1, Uba4 ensures that the adenylation of the C-terminus of Urm1 by the AD precedes the sulfur transfer by the RHD. While the AD

◀

**Figure 5.  The unconventional Uba4–Urm1 reaction scheme and its comparison to canonical UBLs.**

A   Working model of the Uba4-catalyzed reaction cycle of Urm1 thiocarboxylation. After adenylation by the Uba4 AD (ochre), Urm1 (red) forms a thioester intermediate with the catalytic cysteine (C202) located within the crossover loop. The RHD (blue) connected to the AD by a flexible linker reaches Urm1 and replaces the thioester linkage by an acyl-disulfide bond by a nucleophilic attack of the persulfide on the active cysteine (C397). Thiocarboxylated Urm1 is released and serves as sulfur donor for tRNA thiolation.

B   Structural comparison of various E1-UBL system focusing on the position of their catalytic cysteine residues, crossover loops, and nucleotides. Distances between C-terminal glycine (red) and catalytic cysteine residues are indicated unless they are covalently linked. PDB identifiers, nucleotide types, and residue numbers are labeled.

appears available irrespective of different Uba4 conformations, the enzymatic activity of the RHD appears to be silenced in the unbound state by effective shielding of the sulfur-accepting cysteine in the dimeric form of the RHD. Since the RHD of human MOCS3 directly forms monomers in solution (Fig 1C), the regulatory potential of the RHD dimer formation might differ among organisms. Importantly, our data further establish that even after the ATP-dependent binding to Uba4, Urm1 remains bound in an almost identical position during adenylation, thioester formation, and AMP release. This suggests that Urm1 remains in the same location during the final thiocarboxylation reaction as well, since Urm1 binding triggers the release of the flexible linker region and hence the movement of the RHDs. Therefore, the active site of the RHD rather moves into proximity of the C-terminus, than Urm1 shuttling to the RHD domain (Fig 5A). After successful activation by Uba4, Urm1-COSH is released and can be utilized as a sulfur donor for $U_{34}$ thiolation by specific tRNA modification enzymes or participate in ubiquitin-like conjugation reactions.

Urm1 combines the critical features of UBLs and SCPs and likely represents the genuine ancestor of all eukaryotic UBLs like ubiquitin, SUMO, Nedd8, or UFM1 (Xu *et al*, 2006; Leidel *et al*, 2009; Wang *et al*, 2011; Cappadocia & Lima, 2018). The prokaryotic orthologues of the Uba4 AD, ThiF and MoeB, are responsible for the biosynthesis of thiamin and molybdopterin, respectively. Their targets, ThiS and MoaD, are simply activated by adenylation, whereas eukaryotic UBLs require the subsequent formation of a thioester with their respective E1 enzyme (Taylor *et al*, 1998; Leimkühler *et al*, 2001; Duda *et al*, 2005). Like Urm1, Uba4 shares common features and functionalities with both, its simpler prokaryotic and more complex eukaryotic homologues. Therefore, a key to understand the evolutionary branch point when UBLs evolved from SCPs lies in the molecular characterization of Urm1, Uba4, and their concerted reaction cycle.

Our data allow the detailed structural and functional comparison between the eukaryotic Uba4–Urm1 complex and all other available E1-UBL and SCP proteins for the first time. Strikingly, our findings go beyond a simple comparison and indicate an ancestral molecular signature derived from the Uba4–Urm1 system that is still present in all current UBL systems. In detail, we conclude that the intermediate thioester linkage between Urm1 and Cys202 is strictly required to ensure the directed formation of Urm1-COSH without a side reaction that might lead to a detrimental off-target conjugation of Urm1 to Uba4 (Fig EV5F). Since our structural analyses highlight the strong similarities between the Uba4–Urm1 system and all other well-characterized E1-UBL systems (Fig 5B; Olsen *et al*, 2010; Oweis *et al*, 2016; Noda *et al*, 2011; Lake *et al*, 2001; Lee & Schindelin, 2008; Walden *et al*, 2003), it is an exciting possibility that the evolution and diversification of E1 and E2 enzymes for ubiquitin and various UBLs has been shaped and guided by this molecular bottleneck. Despite the fact, that other known E1 enzymes neither harbor an

RHD, nor a linker nor produce thiocarboxylated C-termini. Therefore, the activation of the C-terminus of all these UBLs might occur via a precisely positioned thioester because of their common ancestry from Urm1, before they further diverged in terms of function, complexity, and specificity. Of note, the active-site Cys202 of Uba4 in our substrate-bound structure adopts an almost identical arrangement as Cys173 of Uba2 (the E1 for SUMO) in the tetrahedral intermediate trapped by AVSN (Olsen *et al*, 2010; Fig EV4). Therefore, the Uba4$_{C202K}$-Urm1 structure represents a unique previously uncharacterized reaction step, mimicking the thioester formation after the release of AMP (Fig 5B) that can serve as a model for this reaction intermediate in all other E1-UBL and SCP systems.

In summary, our data provide a comprehensive picture of the molecular characteristics that permit Uba4 to activate Urm1 the SCP for tRNA thiolation and at the same time act as the original prototype for the subsequent development of eukaryotic ubiquitin-conjugation systems.

# Materials and Methods

### Reagents

ATP (NU-1010) was purchased from Jena Bioscience. ADP (A2754), AMP (01930), GTP (G8877), AMP-PNP (10102547001), AlF$_3$ (449628), *tert*-butyl hydroperoxide (458139), hydrogen peroxide (H1009), diamide (D3648), hydroxylamine solution 50 wt. % in H$_2$O (467804), and ammonium sulfide (A1952) were purchased from Sigma-Aldrich. Tris (2-carboxyethyl) phosphine hydrochloride (TCEP) (51805-45-9) was purchased from GoldBio and 1,4-dithiothreitol (DTT001) as well as iodoacetamide (IOD500) from Lab Empire. Chymotrypsin, Sequencing Grade (V1061), was purchased from Promega. ([*N*-Acryloyl-amino]phenyl)mercuric chloride (APM) was synthesized according to (Igloi, 1988).

### Plasmids and mutagenesis

The sequences of human MOCS3 (UniProt ID: O95396), RHD (335–460), *Chaetomium thermophilum* (*Ct*) Uba4 (UniProt ID: G0SC54), and *Ct*Urm1 (UniProt ID: G0SE11) were codon-optimized for expression in *Escherichia coli*, obtained from Genscript, and cloned into a series of modified pET-24d(+) vectors (Novagen/Merck) with various tags using NcoI and KpnI restriction sites (https://www.embl.de/pepcore/pepcore_services/strains_vectors/vectors/bacterial_expression_vectors/popup_bacterial_expression_vectors/). *Ct*Uba4 full length (aa 1–444) was expressed from pETM30 (N-terminal His$_6$ and GST tag) and pETM11 (N-terminal His$_6$ tag) vectors. *Ct*Urm1, *Ct*Uba4 AD (1–290), and *Ct*Uba4 RHD (319–444) were expressed from a pETM30 vector. Human MOCS3 RHD (335–460) was expressed from a

pETM11 vector. The tags are cleavable with tobacco etch virus protease (TEV). Mutations in *Ct*Uba4 and *Ct*Urm1 were introduced using site-directed mutagenesis. In *Ct*Uba4$_{306GGS314}$, residues 306–314 (Gly-Asn-Met-Thr-Gln-Ser-Thr-Asn-Leu) were replaced with three Gly-Gly-Ser repeats. In order to produce thiocarboxylated *Ct*Urm1, *Ct*Urm1 C55S was cloned into pTYB1 (IMPACT™ system, NEB) using NdeI and SapI restriction sites, and subsequently subcloned using NdeI and KpnI into a modified (C-terminal His$_6$ tag) pET30 vector (Novagen). We removed a non-conserved surface cysteine in *Ct*Urm1 to circumvent artifacts during the detection of the thiocarboxylated C-terminus using [(N-acryloylamino)phenyl] mercuric chloride (APM) containing SDS–PAGE.

## Protein sample preparation

WT, mutant, and truncated *Ct*Uba4 proteins were expressed in BL21 (DE3) pRARE in LB media at 20°C using overnight induction with 0.5 M IPTG. For *Ct*Urm1, the induction was carried out at 37°C for 4 h. Bacterial pellets were resuspended in lysis buffer (30 mM Tris–HCl pH 7.5; 300 mM NaCl; 20 mM imidazole; 0.15% TX-100; 10 mM MgSO$_4$; 1 mM β-mercaptoethanol; 10 mg/ml DNase; 1 mg/ml lysozyme; 10% glycerol, and a cocktail of protease inhibitors) and lysed to homogeneity using a high-pressure homogenizer Emulsiflex C3 (Avestin). The proteins were purified with Ni-NTA agarose (Qiagen) under standard conditions. Tags were cleaved with TEV protease and removed with a second Ni-NTA purification step. Subsequently, the proteins were purified by size-exclusion chromatography (SEC) on a HiLoad 26/600 Superdex 200 (*Ct*Uba4 WT and mutant proteins) and HiLoad 16/600 Superdex 75 (*Ct*Urm1 and *Ct*Uba4/human MOCS3 RHD proteins) prep grade columns (GE Healthcare) using ÄKTA™ start. Purified proteins were stored at −80°C in storage buffer (20 mM Tris pH 7.5; 150 mM NaCl and 1 mM DTT).

To obtain thiocarboxylated *Ct*Urm1, the *Ct*Urm1-Intein-CBD-His$_6$ fusion protein was overexpressed in *E. coli* and purified according to published protocols (Kinsland *et al*, 1998; Termathe & Leidel, 2018) with modifications. In brief, the bacterial pellet was resuspended in lysis buffer without reducing agent and lysed to homogeneity. The lysate was passed through a Ni-NTA column, and, following washes, the fusion protein was eluted with elution buffer (30 mM Tris–HCl pH 7.5; 300 mM NaCl; 250 mM imidazole and 10% glycerol). The eluates were dialyzed overnight to chitin-column buffer (30 mM Tris–HCl pH 8 and 500 mM NaCl) and applied on a chitin column. The column was washed with chitin-column buffer, and the cleavage of the tag was induced through incubation with cleavage buffer (30 mM Tris–HCl pH 8; 500 mM NaCl and 35 mM ammonium sulfide) for 16 h at 4°C. This procedure leads to the formation of *Ct*Urm1 without additional residues at the N-terminus and with a thiocarboxylated C-terminal glycine (*Ct*Urm1-COSH). The eluted *Ct*Urm1-COSH was further purified by size-exclusion chromatography on a HiLoad 16/600 Superdex 75 column on ÄKTA™ start system and stored at −80°C in storage buffer (20 mM Tris pH 8 and 150 mM NaCl). The presence of thiocarboxylated C-terminus was confirmed by running the protein on a polyacrylamide gel containing APM. In order to evaluate sample quality, 125 μg of purified proteins was applied on Superdex 75 10/300 GL analytical column (GE Healthcare) using ÄKTA™ pure system and the UV absorbance profiles at 280 and 255 nm were recorded.

## Crystallization, data collection, structure determination, and refinement

For crystallization, *Ct*Uba4 was concentrated to 20 mg/ml in 20 mM Tris pH 7.5; 150 mM NaCl and 1 mM DTT. Crystals of the protein were grown at 21°C by vapor diffusion in sitting drops composed of equal volumes (0.5 μl each) of protein solution and crystallization buffer (50 mM HEPES pH 7.0 and 22% (w/v) PEG 4000). Initial crystals were used to create a seed stock using Seed Beads™ (Hampton Research) according to the classical method provided by the manufacturer. The seed stock was prepared and diluted in stabilizing solution (50 mM HEPES pH 7.0 and 22% (w/v) PEG 4000). In seeding experiments, crystals of Uba4 were grown at 21°C by vapor diffusion in hanging drops composed of equal volumes (1 μl each) of seeds diluted in stabilizing buffer and 5–15 mg/ml of protein in 20 mM Tris pH 7.5; 150 mM NaCl and 1 mM DTT with EDTA pH 8.0 supplemented to a final concentration of 10 mM. Crystals collected from reservoirs containing 50 mM HEPES pH 6.9 and 23.5% (w/v) PEG 4000 were cryoprotected by serial transfer into cryoprotecting buffer (50 mM HEPES pH 7.0; 22% (w/v) PEG 4000 and 30% glycerol). X-ray diffraction data at 2.2 Å resolution were recorded at PETRA III beamline at DESY in Hamburg, Germany.

For stable complex formation, *Ct*Uba4$_{C202K}$, *Ct*Urm1, and ATP (1.00:1.25:2.25 ratio) were incubated in reaction buffer (100 mM MES pH 6.0; 100 mM NaCl and 2 mM MgCl$_2$) at 37°C for 30 min and loaded on Superdex 200 Increase 10/300 GL column on ÄKTA™ Pure system. The fractions containing Uba4–Urm1 were pooled and concentrated to 30 mg/ml. Crystals of *Ct*Uba4$_{C202K}$-*Ct*Urm1 complex were grown at 21°C using hanging drop vapor diffusion technique. In each drop, 2 μl of protein sample was mixed with 2 μl of reservoir solution (0.1 M HEPES pH 7.5 and 20% (w/v) PEG 3350). The crystals appeared after 24 h and grew to maximal size of approximately 50 μm × 150 μm × 250 μm for 1 week. The crystals were cryoprotected by serial transfer into cryoprotecting buffer (0.1 M HEPES pH 7.5; 20% (*w/v*) PEG 3350 and 30% (*v/v*) glycerol) and snap-frozen in liquid nitrogen. X-ray diffraction data at 3.15 Å resolution were collected at XRD2 beamline at Elettra Sincrotrone s.c. source in Trieste, Italy. For data collection details, see Table 1. The structures of *Ct*Uba4 and *Ct*Urm1-*Ct*Uba4$_{C202K}$ complex were determined by molecular replacement using Phaser (McCoy *et al*, 2007) with *E. coli* MoeB (1JW3) and *Ct*Uba4, respectively. Structures were refined using Phenix (Adams *et al*, 2010) and Refmac5 (Winn *et al*, 2011) programs. The "Anisoscale" anisotropy server (Strong *et al*, 2006) has been used to detect and correct mild anisotropy of the datasets. We used map sharpening algorithms implemented in Refmac5 to aid model building of the *Ct*Uba4$_{C202K}$-*Ct*Urm1 structure, which was performed using Coot (Emsley *et al*, 2010). The comprehensive validation was done by MolProbity (Davis *et al*, 2007). Structural visualization was done with PyMOL (http://pymol.sourceforge.net/). For structure refinement statistics, see Table 1.

## Static light scattering

Low-angle light scattering (LALS) and right-angle light scattering (RALS) were measured with a Malvern OMNISEC Reveal instrument (Malvern) connected downstream to an ÄKTA™ purifier equipped with analytical size-exclusion columns. For apo and *Ct*Urm1-bound

*Ct*Uba4 WT and C202S+C397S, 1 mg/ml samples were injected on a Superdex 200 Increase 10/300 GL column (GE Healthcare) and separated in 100 mM MES pH 6.0; 100 mM NaCl and 2 mM MgCl$_2$ at a flow rate of 0.6 ml/min. For *Ct*Uba4 AD, *Ct*Uba4 RHD, and *Ct*Urm1, 1 mg/ml samples were applied on a Superdex 75 10/300 GL column (GE Healthcare) and separated in 20 mM Tris pH 7.5; 150 mM NaCl and 1 mM DTT at a flow rate of 0.25 ml/min. Elution profiles were collected using UV detection at 280 and 255 nm, and LALS, RALS, and refractive index (RI) detection. The molecular weights of elution peaks were calculated using OmniSEC software version 10.40 (Malvern). As calibration standards, 1 mg/ml conalbumin, MW 76 kDa (for Uba4 and Uba4–Urm1 complex), and 1 mg/ml ovalbumin, MW 43 kDa (for remaining proteins) was used and the change in refractive index with respect to concentration was set to 0.185 ml/g (Wen *et al*, 1996).

### Thermal shift assay

The influence of point mutations on thermostability of *Ct*Uba4 protein was measured by thermal shift assay. 10 µg of *Ct*Uba4 was suspended in 20 mM Tris–HCl pH 7.5; 150 mM NaCl; 2 mM DTT; and 2 mM MgCl$_2$ buffer. Each protein was measured in a buffer without addition of nucleotide, with 1 mM AMP or 1 mM ATP. Samples were gradually warmed up from 4°C to 98°C for 2 h. The denaturation process was tracked using the hydrophobic fluorescent dye SYPRO Orange (Sigma Aldrich). Fluorescence was measured using a Bio-Rad CFX96 thermocycler. Melting temperatures were determined from peaks of the first derivative. Data were obtained in three independent experiments with at least two technical replicates.

### GST pull-down assay

WT or mutant *Ct*Uba4 proteins were incubated with GST-*Ct*Urm1 or GST alone in the presence or absence of 1 mM of indicated nucleotides or nucleotide derivatives in reaction buffer (100 mM MES pH 6.0; 100 mM NaCl and 2 mM MgCl$_2$) for 30 min at 37°C. The reaction mix was then added to equilibrated Glutathione Sepharose 4B beads (GE Healthcare) and incubated on a rotating wheel for 90 min at room temperature. After binding, glutathione beads were collected by gentle spinning (500 *g*) and subsequently washed three times with a 0.05% (*v/v*) Tween 20-containing buffer. Bound proteins were denatured at 95°C in the presence of Laemmli sample buffer and analyzed on Bolt™ 4–12% Bis-Tris Plus Gels (Thermo Fisher Scientific). For protein visualization, the gels were stained with Coomassie Brilliant Blue. Inputs were collected after the reaction and before the pull-down and represent 7% of the pull down.

### Pyrophosphate detection assay

ATP hydrolysis capacity of *Ct*Uba4 WT and mutants in the presence of *Ct*Urm1 protein was measured using the fluorometric pyrophosphate assay kit (Abcam). The reactions were performed in a buffer composed of 100 mM MES pH 6.0; 100 mM NaCl; 0.5 mM MgCl$_2$; and 20 µM ATP at 37°C for 30 min. Final *Ct*Uba4 and *Ct*Urm1 concentrations were 5 and 7.5 µM, respectively. Reaction mixtures were centrifuged in 30K Amicon Ultra-0.5 centrifugal filter devices

at 4°C for 30 min at 19,000 *g* in order to separate the pyrophosphate molecules from the protein components. Flow-through samples were used to measure pyrophosphate concentrations, according to manufacturer's guidelines. The fluorescence was measured using a SpectraMax® Gemini EM Microplate Reader (Molecular Devices). Three to five independent experiments were performed, each with two technical replicates.

### Malachite green phosphate assay

ATP hydrolysis capacity of *Ct*Uba4 WT and mutants in the presence of *Ct*Urm1 protein was measured using the Malachite Green Phosphate Assay Kit (Sigma-Aldrich). The reactions were performed in a buffer composed of 100 mM MES pH 6.0; 100 mM NaCl; 2 mM MgCl$_2$; and 160 µM ATP at 37°C for 90 min. Final *Ct*Uba4 and *Ct*Urm1 concentrations were 40 and 60 µM, respectively. The reaction was conducted in the presence of inorganic pyrophosphatase (Thermo Fisher Scientific) in order to catalyzes the hydrolysis of inorganic pyrophosphate generated by Uba4 into phosphate molecules. After the incubation, reaction mixtures were diluted 20 times with water and the orthophosphate concentrations were measured according to manufacturer's instructions. The absorbance was measured using a SpectraMax® 190 Microplate Reader (Molecular Devices). Three independent experiments were performed, each with two technical replicates.

### Uba4–Urm1 adenylate complex formation and detection using SEC

The formation of complexes between *Ct*Uba4 WT/mutant and *Ct*Urm1 in the presence of ATP was monitored by size-exclusion chromatography. The reactions were performed in a buffer composed of 100 mM MES pH 6.0; 100 mM NaCl; 2 mM MgCl$_2$; and 40 µM ATP at 37°C for 30 min. Final *Ct*Uba4 and *Ct*Urm1 concentrations were 20 and 30 µM, respectively. After the reaction, the mixtures were snap-frozen in liquid nitrogen and stored at −80°C. After thawing, the reaction mixtures were applied on a Superdex 200 Increase 10/300 GL column on ÄKTA™ pure system and the integrated peak areas corresponding to ATP (at 255 nm) and *Ct*Urm1 (at 280 nm) were calculated using the UNICORN 7.0 software. The fractions containing the *Ct*Uba4-*Ct*Urm1 complex were run on SDS–PAGE gels.

### Urm1 conjugation assay

20 µM of *Ct*Uba4 and 20 µM of carboxylated or thiocarboxylated *Ct*Urm1 C55S were mixed in reaction buffer (20 mM Tris pH 8.0 and 200 mM NaCl). 2.5 mM *tert*-Butyl hydroperoxide, 2.5 mM H$_2$O$_2$, 2.5 mM diamide, and 2.5 mM ATP were included and excluded as indicated. The reaction mix was incubated for 30 min at 37°C, stopped by adding Laemmli sample buffer containing DTT, and incubated for 5 min at 95°C. TCEP, DTT, and hydroxylamine were added at a final concentration of 5 mM after the initial 30-min reaction and further incubated for 5 min at 37°C before Laemmli sample buffer addition. Subsequently, the samples were loaded on Bolt™ 4–12% Bis-Tris Plus Gels (Thermo Fisher Scientific). For protein visualization, the gels were stained with Coomassie Brilliant Blue.

## Mass spectrometry analysis of Urm1 conjugation site

The gel bands were cut out and destained by washing four times alternately in 25 and 50% acetonitrile in 25 mM $NH_4HCO_3$. Proteins were reduced with 50 mM DTT at 37°C for 45 min and alkylated with 55 mM iodoacetamide at room temperature for 2 h in the dark. Residual reagents were removed with 50% acetonitrile in 25 mM $NH_4HCO_3$. The bands were dehydrated in 100% acetonitrile, dried, and rehydrated in 25 μl of chymotrypsin solution (10 ng/μl; 25 mM $NH_4HCO_3$ pH 8.0 and 10 mM $CaCl_2$). After 15 min, additional 40 μl of 25 mM $NH_4HCO_3$ was added. After overnight incubation at 25°C, the digestion was stopped by adding 0.5% trifluoroacetic acid. Peptides were extracted by sonication and drying with 100% acetonitrile. The extracts were evaporated to dryness and resuspended in 2% acetonitrile with 0.05% trifluoroacetic acid. Samples were analyzed with a Q-Exactive mass spectrometer (Thermo Fisher Scientific) coupled with a nanoHPLC (UltiMate 3000 RSLCnano System, Thermo Fisher Scientific). Peptides were loaded onto a trap column (Acclaim PepMap 100 C18, 75 μm × 20 mm, 3 μm particle, 100 Å pore size, Thermo Fisher Scientific) in 2% acetonitrile with 0.05% TFA at a flow rate of 5 μl/min and further separated on analytical column (Acclaim PepMap RSLC C18, 75 μm × 500 mm, 2 μm particle, 100 Å pore size, Thermo Fisher Scientific) at 50°C with a 60-min gradient from 2 to 40% acetonitrile in 0.05% formic acid at a flow rate of 200 nl/min. The eluted peptides were ionized using a Digital PicoView 550 nanospray source (New Objective). The Q-Exactive was operated in a data-dependent mode using the top eight method with 35 s of dynamic exclusion. Full-scan MS spectra were acquired with a resolution of 70,000 at $m/z$ 200 with an automatic gain control (AGC) target of 1e6. The MS/MS spectra were acquired with a resolution of 35,000 at $m/z$ 200 with an AGC target of 3e6. The maximum ion accumulation times for the full MS and the MS/MS scans were 120 and 110 ms, respectively. The lock mass option was enabled for survey scans to improve mass accuracy. The raw files acquired by the MS system were processed using the Proteome Discoverer platform (v.1.4; Thermo Scientific). Obtained peak lists were searched using an in-house MASCOT server (v.2.5.1; Matrix Science, London, UK) against the cRAP database (https://www.thegpm.org/crap/, released August 2019) with manually added $CtUba4_{C202K}$, $CtUba4_{C202S}$, and $CtUrm1$ protein sequences (122 entries in total). The following search parameters were applied: chymotrypsin enzyme specificity (C-term of ADEFLMWY); up to five missed cleavages allowed; peptide mass tolerance: 10 ppm; fragment mass tolerance: 20 mmu; fixed modification: carbamidomethyl (C); and variable modifications: oxidation (M) and Gly-Gly-His (K), which is a remnant motif of Urm1 generated after chymotrypsin digestion. MS/MS spectra of peptides containing modified lysine were manually inspected, and identifications with Mascot score higher than 40 were considered reliable. For $Uba4_{C202S}$-Urm1, data were obtained in two independent experiments with two technical replicates.

## Generation of yeast strains

Endogenously tagged strains were generated using a standard procedure of a plasmid-based strategy to exchange the G418 cassette of the $uba4\Delta$ strain with the tagged version of Uba4 mutant genes. For all mutant strains, at least two independent clones were generated and tested. Yeast cultures were grown overnight at 30°C in an appropriate selection medium. Pre-cultures were diluted to an $OD_{600}$ of 0.1 in YPD and further grown till an $OD_{600}$ between 0.8 and 1. Aliquots were taken for further analysis, harvested by centrifugation, and snap-frozen. $OD_{600}$ of cultures was adjusted to 0.75, and a 1:5 dilution series was spotted on YPD supplemented with 3 nM rapamycin. Plates were incubated for 3 days at 37°C prior to imaging. All yeast strains used are listed in Appendix Table S3.

## RNA isolation and Northern blot analysis

Aliquots from spotting experiment were thawed on ice, and total RNA was extracted by using hot phenol/chloroform extraction. 1 μg of total RNA was resolved on an 8% PAGE containing 0.5× TBE, 7 M urea, and 50 μg/ml APM (Igloi, 1988). Northern blot analysis was performed as described previously by using the probe against $tRNA_{UUC}^{Glu}$ (5′-tggctccgatacgggggagtcgaac-3′; Leidel et al, 2009).

## Protein isolation and Western blot analysis

Aliquots (3 $OD_{600}$ units) from spotting experiment were thawed on ice, and total proteins were extracted as previously described (von der Haar, 2007). Total protein extracts were resolved by SDS–PAGE and transferred by semi-dry blotting onto a PVDF membrane. Membranes were probed using a monoclonal anti-HA antibody (Covance MMS-101R).

# Data availability

The atomic coordinates and respective structure factors for full-length CtUba4 (PDB ID 6YUB; http://www.rcsb.org/pdb/explore/explore.do?structureId=6YUB) and the CtUba4C202K-Urm1 complex (PDB ID 6YUC; http://www.rcsb.org/pdb/explore/explore.do?structureId=6YUC) have been validated and deposited at the European Protein Data Bank (http://www.wwpdb.org). The related entry (PDB ID 6Z6S; http://www.rcsb.org/pdb/explore/explore.do?structureId=6Z6S) contains the structure factors before anisotropy correction. The mass spectrometry data have been deposited to the ProteomeXchange Consortium via the PRIDE (Perez-Riverol et al, 2019) partner repository with the dataset identifier PXD015802; http://www.ebi.ac.uk/pride/archive/projects/PXD015802.

**Expanded View** for this article is available online.

## Acknowledgements

We thank the members of the Glatt and Leidel laboratories for discussion and suggestions and Kay Hofmann, Yogesh Kulathu, and Thimo Kurz for feedback on the manuscript. Furthermore, we thank the beamline staff at beamlines 14.1 (BESSY), XDR2 (Elettra Sincrotrone Trieste), and P11 (PETRAIII) for support during data collection and the Heddle and Kantyka labs at the MCB for the access to specialized equipment. This work was supported by the First Team Grant (FirstTEAM/2016-1/2; KER, MS, SG) from the Foundation for Polish Science and the OPUS16 grant (2018/31/B/NZ1/03559; RK and SG) from the National Science Centre, by the NCCR RNA & Disease, funded by the Swiss National Science Foundation (SDK and SL) and the Max Planck Society (MT, SDK, and SL). In addition, we thank the MCB structural biology core facility (supported by the TEAM TECH CORE FACILITY/2017-4/6 grant from Foundation

for Polish Science) for providing instruments and support, in particular, Klaudia Woś for her efforts and constant assistance during crystallization trials. This project has received funding from the European Union's Horizon 2020 research and innovation program under grant agreement No 730872. The open-access publication of this article was funded by the Priority Research Area BioS under the program "Initiative of Excellence—Research University" at the Jagiellonian University in Krakow.

## Author contributions

MP, MT, SAL, and SG envisioned the project and designed the experimental concepts. MP and KER performed biochemical, biophysical, and crystallographic analyses; MP and KER purified proteins and characterized mutants using biophysical analyses with the help of RK and MS; MT and SDK generated and characterized yeast mutants and performed *in vivo* and *in vitro* analyses. MP, KER, PG, and SG collected crystallographic data, refined structures, and analyzed the crystallographic results. UJ performed mass spec analyses with the support of KER. MP, RK, MT, SDK, KER, SAL, and SG prepared figures and wrote the manuscript, with the input from all other authors.

## Conflict of interest

The authors declare that they have no conflict of interest.

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
