## [Review Process File · The EMBO Journal]

Molecular basis for the bifunctional Uba4-Urm1 sulfur relay system in tRNA thiolation and ubiquitin-like conjugation

Marta Pabis, Martin Termathe, Keertiraju Ravichandran, Sandra Kienast, Rościsław Krutyhołowa, Mikołaj Sokołowski, Urszula Jankowska, Przemysław Grudnik, Sebastian Leidel, and Sebastian Glatt

DOI: [10.15252/embj.2020105087](https://doi.org/10.15252/embj.2020105087)

Corresponding author(s): Sebastian Glatt (sebastian.glatt@uj.edu.pl), Sebastian Leidel (sebastian.leidel@dcb.unibe.ch)

Review Timeline:

Submission Date:	24th Mar 20
Editorial Decision:	14th Apr 20
Revision Received:	29th May 20
Editorial Decision:	17th Jun 20
Revision Received:	23rd Jun 20
Accepted:	26th Jun 20

Editor: Hartmut Vodermaier

Transaction Report:

Dr. Sebastian Glatt
Jagiellonian University
Malopolska Centre of Biotechnology
Gronostajowa 7a str
Krakow 30-387
Poland

14th Apr 2020

Re: EMBOJ-2020-105087

Molecular basis for the bifunctional Uba4-Urm1 sulfur relay system in tRNA thiolation and ubiquitin-like conjugation

Thank you for submitting your manuscript on Uba4-Urm1 structures and mechanism for our editorial consideration. I have now received reports from three expert reviewers, copied below for your information.

Since all referees acknowledge the interest and potential importance of this work, we would in principle be interested in considering a revised manuscript further for EMBO Journal publication. As you will see, the referees nevertheless also raise a number of significant concerns that would need to be decisively addressed before publication would be justified. This includes presentational issues as well as queries regarding methodologies and structural analyses; however, I am afraid there are also concerns whose clarification will require additional experimental work, in particular the possibility of oxyester bond formation in the C202S mutant, and the possible stabilization of the Uba4-Urm1 complex via a covalent bond in the pulldown experiment. The importance of such revision experiments was reinforced during the post-review discussions between the referees.

REFEREE REPORTS

Referee #1:

Urm1 is a ubiquitin-like protein that is adenylated and thiocarboxylated by Uba4, an E1-like

activating enzyme. Thiocarboxylated Urm1 has two roles, one is serving as a sulfur donor for specific tRNA thiolases and another is modifying proteins similarly with ubiquitin and other UbIs. In this point, Uba4 and Urm1 are located at an evolutionary branch point between UBL and sulfur-relay systems. However, their structural and functional details have remained elusive. Pabis et al. determined the crystal structures of apo Uba4 and Uba4-Urm1 complex and provided the first structural basis of this unique E1-like enzyme. Uba4 is a homodimer and each protomer is composed of AD, RHD and their flexible linker. Interestingly, the authors showed that in the apo structure, Uba4 forms an asymmetric homodimer with RHD-RHD interaction, whereas in the Urm1-bound structure, Urm1 displaces the linker and RHD, which leads to the loss of RHD-RHD interaction and formation of symmetric Uba4 dimer. Moreover, the authors performed structure-based mutational analyses and proposed the mechanism of the reaction cycle mediated by Uba4 and the protection mechanism of self-conjugation.

Reported structures are novel and give us lots of information about this unique E1-like enzyme. The authors also performed lots of functional analyses based on the structure and proposed the unconventional Uba4-Urm1 reaction scheme. However, this manuscript possesses some critical shortcomings listed below, which must be resolved prior to be published at EMBO Journal.

Major points

1) The authors analyzed the formation of covalent-bond between Uba4 and Urm1 by SDS-PAGE (Figure 4A), in which Uba4 C202S formed covalent-bond with Urm1 much more efficiently than Uba4 WT. Using MS analysis, the authors showed that Urm1 formed covalent-bond with Lys161, Lys191, and Lys192 of Uba4 C202S. Based on these results, the authors proposed that C202 protects against covalent attachment of Urm1 to these lysines. However, it was not shown that the Uba4-Urm1 band in Figure 4A corresponds to Urm1 attached to lysines in Uba4. It is known that when catalytic cysteine is replaced with Ser, ester bond instead of thioester bond is formed between E2 and Ubl and that ester bond is much more stable than thioester band and could be analyzed by SDS-PAGE. Similarly, when C202 is replaced with Ser, S202 could form ester bond with Urm1, which might be a major component in the Uba4-Urm1 band in Figure 4A. Perform similar experiments using Uba4 C202A mutant. If Uba4(C202A)-Urm1 band appeared similarly with Uba4(C202S)-Urm1, the authors can claim that C202 protects against covalent attachment.

2) The authors showed that ATP and AMP-PNP markedly increased the binding affinity of Urm1 to Uba4 while mild and no increase were observed for ADP and AMP, respectively (Figure 2A). These observations suggest that gamma-phosphate group is important for increasing the affinity and beta-phosphate group also contributes to that to some extent. Prepare ATP-bound model of Uba4-Urm1 complex and discuss the mechanism of ATP-mediated affinity enhancement based on the structural model.

3) In Table 1, Rmeas and $1/\sigma$ values for the outer shell of CtUba4-CtUrm1 are too bad. Re-perform crystallographic refinement using lower resolution data ($1/\sigma$ value for outer shell must be larger than 1.0). Provide CC1/2 value for outer shell in Table 1.

4) In Table 1, B-factors for proteins in the complex are too large (189). Is there any reason for that? Provide a structural model colored with B-factors and non-biased omit map of the Urm1 C-terminus.

Minor points

1) In page 6, lines 14-16, "We found that ... This indicates that hydrolysis occurs between the alpha- and beta-phosphate", This logic is difficult to understand. Why could authors know the hydrolysis site based on the affinity difference? It may be possible that gamma-phosphate of ATP interacted with Urm1, which increased the affinity between Uba4-Urm1 (related to Major point 2).

- 2) In page 6, lines 18-19, "while AMP increased... indicating the formation of a stable adenylate intermediate", This sentence is confusing. E1 enzymes catalyze adenylation of Ubls using ATP and NOT AMP. It is impossible to form an adenylate intermediate of Urm1 using AMP.
- 3) In page 7, line 9, "a strong hydrogen network" should be "a strong hydrogen-bond network".
- 4) In page 7, line 23, remove "site" after (C208A/C211A).
- 5) In Table 1, provide unit for B-factors.

Referee #2:

The covalent modification of target proteins with ubiquitin or ubiquitin-like modifiers is initiated by ubiquitin/ubiquitin-like activating enzymes also referred to as E1 enzymes. These enzymes have evolved from simpler ancestors such as the *E. coli* MoeB and ThiF proteins, which catalyze the sulfur-incorporation steps during the biosynthesis of molybdopterin and thiamin, respectively. The fungal Uba4 enzyme and its human counterpart MOCS3 possibly represent a link between these activities since they catalyze a thiolation reaction, namely the conversion of uridin³⁴ to thio-uridine in eukaryotic RNAs, and also the activation of a cognate ubiquitin-like protein, referred to as Urm1. The understanding of how these two activities are accomplished by one enzyme has been far from complete and this is where this manuscript makes an important contribution.

Pabis et al. describe crystal structures of Uba4 from *Chaetomium thermoautotrophicum* in its apo-state and in complex with Urm1. The structures are strengthened by biochemical structure-function studies in vitro but also in *S. cerevisiae* cells. Uba4 consists of an adenylation domain present in all E1 enzymes and also the MoeB/ThiF biosynthetic proteins and a unique rhodanese-like domain which, in its persulfide state, serves as sulfur donor for Urm1 thiocarboxylate formation. The apo-structure revealed an asymmetric arrangement, although both domains independently dimerize, however, the C2 axes are not aligned. In the Uba4-Urm1 complex the rhodanese-like domains become disordered and a C2 symmetric heterotetramer is formed. The linker region connecting the two domains of Uba4 was found to play a critical role during the catalytic mechanism.

Major concerns:

1. Crystallography:

a. The PDB validation reports were incomplete and lacked an independent confirmation of the reported R-factors. Maybe because no experimental data were submitted? If this assumption is correct, why not? Without this information, i.e. an independent conformation of the reported R-factors I cannot endorse publication of this study.

2. Table 2 is missing some key statistics, in particular values for $CC(1/2)$ and $R(pim)$. Without these statistical parameters I suspect that the resolution limit of the complex is too optimistic based on the values for $R(meas)$ and $. CC(1/2)$ and $R(pim)$ will allow the reader to conclude whether the high resolution cutoff makes sense. Since the authors did not mention whether either or both of these crystals diffracted X-rays anisotropically, one would assume that this is not the case. However, if there was significant anisotropy for one or both of the crystal forms, the data need to be reprocessed with the Staraniso server from Global Phasing. Were ncs restraints (obviously domain-wise) employed during the refinement of the apo-structure. Even at 2.2 Å this will most likely improve model quality and possibly also narrow the gap between R and $R(free)$ which for my taste is at the high end of what I would like to see at this resolution.

2. RHD-dimerization

a. Based on the available figures the interface between the AD dimer and RHD dimer seems to be minimal. This raises the question whether the arrangement and, in particular, the asymmetry seen here is a crystallization artifact? What are the packing interactions that the RHD dimer is involved in? How extensive are they, do they involve a symmetry-related AD dimer or RHD dimer or both? If they involve the AD dimer, could these interactions be of relevance also for an interaction in cis?

b. The authors conclude (page 10, lines 7-9) without experimental evidence that the RHD dimer dissociates and in the discussion treat this as an observation and not as a hypothesis. Given the fact that the isolated RHD seems to stably dimerize considerable forces would be required to pull the RHDs apart. Where should that energy come from? It can only come from the binding of Urm1 and the energy would then not only be required for the dissociation of the RHD dimer but also for the displacement of the linker between the AD and RHD. Under normal circumstances I would request experimental verification of this hypothesis, however, as I cannot judge whether it is possible for the authors to carry out experiments at all under the current circumstances, an alternative approach would be to state this as a hypothesis throughout the manuscript and point out that there is no experimental proof. In this context, it would be informative if the authors included the buried surface area upon RHD dimerization.

3. Binding order for ATP and Urm1

Based on thermal shift analyses in which the binding of different nucleotides to Uba4 was investigated the authors ultimately come to the conclusion (page 6, lines 23-25) that "This is in striking contrast to all other known canonical and non-canonical E1 enzymes that bind their UBL in the absence of ATP and Mg(2+)-ions". Old kinetic data (Haas & Rose, JBC, 1982) postulate strictly ordered binding for Uba1 isolated from rabbit reticulocytes with ATP as leading substrate. This is certainly pertinent information in this context and should be mentioned/discussed. The previous sentence (lines 21-23) is trivial and is, of course, also true for every E1 enzyme, so please omit or rephrase. I also object to the statement that Urm1 does not bind to "ATP alone" (line 20) as the authors' own data contradict this conclusion. What the authors observe is a decrease in stabilization as one goes from AMP over ADP to ATP. From the manuscript I am also unclear whether Mg was present in these thermal shift assays, since there is no mention of Mg in the corresponding Materials and Methods section I would conclude that this is not the case, however, under these circumstances the data would be useless and have to be repeated with Mg being present. I am also confused regarding the statement (line 19) that the increased stability of the AMP complex indicates the formation of a "stable adenylate intermediate". The acyladenylate intermediate can only form in the presence of the Urm1 C-terminus.

4. Residue numbers and identities in Ct Urm1 vs. Sc Urm1

While Table S1 finally guides the reader in translating from the Ct residue numbers and identities to those of Sc, this is too late and too cumbersome for the reader. The authors need to come up with a better way to guide the reader through Figs. 2D, 4C and S8E as well as the corresponding sections in the text without having to look at this table over and over again.

Minor points:

1. Page 3, line 20: Replace "multi-domain" with "two-domain".
2. Page 4, lines 15/16: Are there really structures available for all other eukaryotic E1 enzymes? I am pretty sure this is not the case.
3. Page 5, lines 6-8: Please rephrase, e.g. "While the AD and RHDs form C2-symmetrical dimers the structure is asymmetric since the two 2-fold axes do not coincide." or something similar.

4. Page 5, line 19 and whenever this is present in one of the figures: What is an "active loop"? Since the Cys is part of the loop it probably should be called "active site loop".
5. Page 8, line 15: Please state that the nine residues in the linker region were replaced with three Gly-Gly-Ser repeats, at least that was my interpretation.
6. Page 10, lines 3-4: It might be helpful to state that the twofold axis of the Uba4-Urm1 complex is a crystallographic symmetry element.
7. Page 10, line 7: Replace "from one molecule" with "from the one molecule where it is bound"
8. Page 10, lines 7-9: I fail to see how the second sentence is a logical conclusion from what is stated in the first sentence.
9. Page 11, line 25: Are these two cysteines visualized in one of the figures? If yes, reference here, if not, include a suitable figures or modify an existing figure accordingly.
10. Page 12, line 2: "Cys311 is less critical ..." Less critical than what exactly?
11. Page 14, lines 22-24: This part of the discussion would suggest that the reaction catalyzed by Uba4 and the evolutionary ancestors of the E1 family, MoeB and ThiF, are more complex than let's say the E1 for ubiquitin and that these modern E1 have evolved in such a way as to prevent interactions with rhodanases.
12. Legend to Fig. 1
 - (A) Rephrase to "rhodanese-like domains (RHDs) of either Tum1 or Uba4." to reflect the arrows drawn in the figure.
 - (B) Add "RHD: Rhodanese-like domain"
 - (D) "Dotted line" should be "dashed line" and "(middle)" should be replaced with "(central inset)" or something similar.
13. Fig. 2B: Highlight Asn307 like the other residues which have been studied
14. Legend to Fig. 2:
 - (B) Add "(for reference see full structure on upper left)" after "Close up" and after "zinc site" add "(inset, lower left)".
15. Legend to Fig. 3:
 - (C) Insert "which form the dimer" after "adjacent asymmetric unit"
16. Fig. 4D:

Will this be readable, even if shown at higher resolution?
17. Legend to Fig. 5:
 - (B) Add "unless they are covalently linked" after "are indicated."
18. Table 1

"l/sigl" must be ""

I doubt that a precision to 1/1000 Å is warranted for the unit cell dimensions of the apo-structure. Which ligands/ions are present in the apo-structure? With an average B-factor of >187 Å² the question arises, are these ligands really present?
19. Page 29, line 23: Briefly describe why the C55S variant was used.
20. Page 20, line 18: Insert "published protocols" after "according to".
21. Page 32, line 8: Since "maximal size" is mentioned, how big were the crystals?
22. Page 32, line 14: How was Urm1 modeled? By MR or built de novo or positioned as a rigid body in the map?
23. Fig S3A: Use decimal points instead of commas.
24. Legend to Fig. S2
 - (C) Please define "BtTST".
25. Fig.S4B: Label first and last residue of loop as well as residues, which are adjacent to the gap. Is the dashed link representing the disordered residues shifted to the right? It certainly does not connect the residues adjacent to the gap.
26. Fig. S6B

Add gel analysis for the fractions from the Uba4-Urm1 SEC run in the absence of ATP.

27. Fig. S7

(A) Right panel: Instead of showing a crowded superposition containing two maps show them side-by-side.

(B) Replace "Full-length Uba4" with "Apo-Uba4".

Referee #3:

The manuscript "Molecular basis for the 2 bifunctional Uba4-Urm1 sulfur relay system in tRNA thiolation and ubiquitin-like conjugation" by Pabis. et al. reports the crystal structures of the full length Uba4 alone and in complex with Urm1. These structures are complemented with biochemical analysis including structure-based mutations that are tested for their effects on yeast growth. Based on their results the authors suggest that the apo Uba4 forms an asymmetric dimer, and both the AD and RHD contribute to this structure. In this structure, the RHD is inactivated. Moreover, using pull down experiments they show that Uba4 does not bind Urm1 if ATP is absent. Interestingly, the linker connecting the AD to the RHD is visible only in one molecule of Uba4. In that molecule, part of the linker is in close proximity with the ATP binding site. The authors suggest that the linker coordinates the Uba4 catalytic steps starting from ATP binding to thiocarboxylation. Finally, in the structure of Uba4 linked to Urm1 via an isopeptide bond, they show that Uba4-Urm1 forms a symmetric dimer. In this structure the dimerization of the RHD is disrupted. This, in turn, exposes the active site Cys of RHD which is buried in the apo structure.

Taken together, this work advances our understanding of Uba4 activity and provide novel structural mechanism for the crosstalk between the AD and the RHD. However, the data supporting the conclusions require further validation.

Major points:

1. The authors performed pulldown experiments with GST-Urm1 and Uba4 to show that Uba4 binds Urm1 only in the presence of ATP. They also argue that this is unique to Uba4 compared to other E1 enzymes (page 6; 23-25). Based on their results one cannot rule out the possibility that the increase of binding in the presence of ATP is due to the formation of thioester bond between Urm1 and Uba4 active site (Cys202). This stabilizes the interaction between the two proteins. Finally, when the complex is loaded on the gel and breaks down due to reducing agent, the intensity of the Uba4 increases and this leads to the conclusion that ATP increases the affinity between the two proteins. This is not unique to Uba4 and holds true with other E1 enzymes and their Ubls. Therefore, in order to rule out this possibility the authors have to redo the pulldown experiments with Uba4 C202A that cannot form thioester bond with Urm1.
2. The authors suggest the linker contacting the AD to the RHD is critical for the proper functioning of Uba4. Based on the structural data they suggest that it affects ATP binding. These conclusions are based on the structure of the linker in one Uba4 molecule but not in the other. My question is how the authors know that the structured linker in one of the molecules is not due to crystal packing? One possibility is that the mutations in the linker are structural mutations that affect the protein structure, similarly to the mutation in the Zn binding site.
3. It is not clear why the authors decided not test the same mutations both in the in vitro and in vivo assays. For example, the D134A mutation which has strong effect on ATP hydrolysis was not checked in thiolated tRNA or yeast growth. This is only one example. Please add this information.
4. The linker is critical for the catalytic activity of Uba4 (page 8; 17-18). If this is the case please show that the AD domain alone does not hydrolyze ATP.
5. The mutations N307/M308 to Ala stimulates ATP hydrolysis. Is it because the protein is more

stable? and not because it mimics URM1 binding. Also, these residues are not conserved in yeast (table S1)

6. It is not clear why the mutation of C397S but not C202 affects ATP hydrolysis. Following the model (Fig 5), C397 functions after the formation of the thioester bond with C202, so how can C397 affect hydrolysis but not C397?

7. The authors suggest that in the Uba4-urm1, the RHD is no longer a dimer. Moreover, although the RHD is not observed in the structure they claim that C397 of the RHD is getting next to C202. This is too speculative and has to be shown.

8. In Fig 4a why ATP is present only with the WT Uba4 and not with the C202S mutation? Does Urm1 form oxyester bond with Uba4 C202S. What is the effect of the C202A mutation? Does it behave as C202S?

Minor points

1. Figure 1A- it should be AD and not UBA4

2. Figure 1B- why the active site Cys397 of RHD is not shown

3. What is the surface representation in fig 1D? why not to have the AD and AD' at a different color?

4. Please provide the rmsd of the superpositions in Fig 1E

5. Fig2B the label for Asn307 is missing

6. Fig3C please add in the legend the red and green colors.

7. Please add cc1/2 value for the crystallographic table

Point-by-Point Response

EMBOJ-2020-105087

Referee #1:

Urm1 is an ubiquitin-like protein that is adenylated and thiocarboxylated by Uba4, an E1-like activating enzyme. Thiocarboxylated Urm1 has two roles, one is serving as a sulfur donor for specific tRNA thiolases and another is modifying proteins similarly with ubiquitin and other Ubls. In this point, Uba4 and Urm1 are located at an evolutionary branch point between UBL and sulfur-relay systems. However, their structural and functional details have remained elusive. Pabis et al. determined the crystal structures of apo Uba4 and Uba4-Urm1 complex and provided the first structural basis of this unique E1-like enzyme. Uba4 is a homodimer and each protomer is composed of AD, RHD and their flexible linker. Interestingly, the authors showed that in the apo structure, Uba4 forms an asymmetric homodimer with RHD-RHD interaction, whereas in the Urm1-bound structure, Urm1 displaces the linker and RHD, which leads to the loss of RHD-RHD interaction and formation of symmetric Uba4 dimer. Moreover, the authors performed structure-based mutational analyses and proposed the mechanism of the reaction cycle mediated by Uba4 and the protection mechanism of self-conjugation.

Reported structures are novel and give us lots of information about this unique E1-like enzyme. The authors also performed lots of functional analyses based on the structure and proposed the unconventional Uba4-Urm1 reaction scheme. However, this manuscript possesses some critical shortcomings listed below, which must be resolved prior to be published at EMBO Journal.

Major points

1) The authors analyzed the formation of covalent-bond between Uba4 and Urm1 by SDS-PAGE (Figure 4A), in which Uba4 C202S formed covalent-bond with Urm1 much more efficiently than Uba4 WT. Using MS analysis, the authors showed that Urm1 formed covalent-bond with Lys161, Lys191, and Lys192 of Uba4 C202S. Based on these results, the authors proposed that C202 protects against covalent attachment of Urm1 to these lysines. However, it was not shown that the Uba4-Urm1 band in Figure 4A corresponds to Urm1 attached to lysines in Uba4. It is known that when catalytic cysteine is replaced with Ser, an ester bond instead of a thioester bond is formed between E2 and Ubl and that the ester bond is much more stable than a thioester band and could be analyzed by SDS-PAGE. Similarly, when C202 is replaced with Ser, S202 could form ester bond with Urm1, which might be a major component in the Uba4-Urm1 band in Figure 4A. Perform similar experiments using Uba4 C202A mutant. If the Uba4(C202A)-Urm1 band appeared similarly with Uba4(C202S)-Urm1, the authors can claim that C202 protects against covalent attachment.

Response: It is important to distinguish between Urm1 carrying a carboxy (COOH) and thiocarboxy (COSH) group at the C-terminus. Whereas, Urm1-COSH forms these conjugates with Uba4 C202S, Urm1-COOH does not form these conjugates (Figure 4A) even in the presence of oxidative stress. In the initially submitted manuscript, we described experiments showing that the addition of DTT, TCEP and hydroxylamine does not resolve the observed conjugates (Figure 4A). As described in the revised manuscript on page 12 – “...None of these reagents disrupted the conjugates, showing that a covalent iso-peptide bond is formed between Urm1-COSH and Uba4_{C202S}.” We assume that these reagents should also resolve oxy-ester bonds. However, the used experimental approach cannot exclude that the detected lysine conjugates represent only a minor fraction in the sample. Therefore, we agree that the

previously presented results in combination with the mass spec analyses only indirectly show that the observed band represents Urm1-conjugates on the identified lysine residues. Hence, we followed the recommendation of this reviewer to exclude the ester bond and purified the suggested Uba4_{C202A} mutant (also requested by reviewer 3 for other reasons). Strikingly, the Uba4_{C202A} mutant produces the identical conjugates as the previously used Uba4_{C202S} mutant. In addition, we show that these conjugates induced by mild oxidative stress are resistant to Hydroxylamine (HA) treatment. Our additional experiments confirm that Cys202 indeed protects Uba4 against covalent attachment by its own product, Urm1-COSH.

Furthermore, we combined the previously analyzed C202S mutant with substitutions of the three main lysine residues identified by mass spec (Lys161, Lys191 and Lys192), leading to Uba4_{C202S/K161A/K191A/K192A}. In this quadruple mutant, the band of the conjugates almost completely disappears, confirming the lysine dependency of the shifted band and directly validating our mass spec analysis. Of note, we did detect low incidence linkages to Lys124 and Lys225 (Appendix Table S3), which could cause the faint remaining band of conjugates, which are also resistant to HA.

We now present the results of these additional experiments in the newly created Figure 4G and mention the results in the revised manuscript, which now reads as follows. *“Interestingly, we identified a set of covalently linked lysine residues (Lys161, Lys191 and Lys192) in the TBH treated samples of Uba4C202S (Figure 4E) close to the active site (Figure 4F). Furthermore, we show that a variant of C202S mutant lacking the lysine residues C202S/K161A/K191A/K192A shows a strong reduction of the Urm1 conjugates (Figure 4G). Our data confirms that Urm1-COSH is covalently attached in the absence of Cys202 with a low selectivity of the C202S-triggered conjugation reaction. Whereas Uba4C202S and an equivalent Uba4C202A mutant show enhanced conjugation (Figure 4G), substitutions of other active-site or surface cysteines (i.e. C305S, C397S) do not promote the formation of Uba4-Urm1 conjugates (Figure 4B and Figure EV5D, E).”*

In addition, we show that none of the newly introduced mutations affect the stability of the protein and included these data into the revised Appendix Table S1. We have fused the two structural representations of the available and conjugated lysine residues into one panel to accommodate the new data and to simplify the figure layout of Figure 4F. Please note that we had previously used the ScUba4 C225A mutant in all *in vivo* analyses and the inclusion of the data on CtUba4 C202A further increases the consistency between the presented *in vitro* and *in vivo* analyses.

2) The authors showed that ATP and AMP-PNP markedly increased the binding affinity of Urm1 to Uba4 while mild and no increase were observed for ADP and AMP, respectively (Figure 2A). These observations suggest that gamma-phosphate group is important for increasing the affinity and beta-phosphate group also contributes to that to some extent. Prepare ATP-bound model of Uba4-Urm1 complex and discuss the mechanism of ATP-mediated affinity enhancement based on the structural model.

Response: We are confident that the increased complex formation between Urm1 and Uba4 is not caused by the presence of an ATP molecule itself, but by the adenylation of the C-terminus of Urm1 and possibly the formation of a thioester intermediate (see also response to reviewer 3). Using different nucleotides, we confirmed that Uba4 conducts a canonical adenylation reaction, hydrolyzing ATP into AMP and pyrophosphate. We use AMPPNP, which harbors a non-hydrolysable beta-gamma bond, to show that the hydrolysis indeed occurs between alpha- and beta-phosphate, which is still available for hydrolysis in AMPPNP. Of note, we also detect the production of pyrophosphate directly (Figure 2C).

We performed additional experiments and tested the D134A+Q164A and C202A mutants in our GST-pull down assays. We now show that both mutants that display strongly reduced ATP hydrolysis rates, namely R18A+R77A and D134A+Q164A, also show decreased complex formation. Both, the C202S and C202A mutants show normal complex formation, indicating that the adenylation of Urm1 itself stabilizes the complex, rather than the thioester intermediate that cannot be formed by these mutants. As neither AIF3-ADP, ADP, AMP nor GTP promote adenylation of Urm1, they do not stabilize the complex. We attempted to measure the affinity of Uba4 variants to fluorescently labeled MANT-AMPPNP, but due to steric hindrance the labeled nucleotides did not fit into the binding pocket. In the absence of quantitative data for ATP binding, it is difficult to confirm that R18A+R77A and D134A+Q164A are still able to bind the nucleotide and discriminate between the effect of ATP binding and adenylation itself.

The new data directly comparing complex formation of different mutants in the absence and presence of ATP is now available in Figure EV2D. Accordingly, we added a dedicated paragraph for this topic on page 9 to clarify the interpretation of our data, which now reads as follows in the revised version – *“Finally, we used the set of Uba4 mutants to identify the specific reaction intermediate that stabilizes the Uba4-Urm1 complex after addition of ATP and AMP-PNP (Figure 2A). The previous observation of a hydrolysis event between the α - and β -phosphate and the production of pyrophosphate corroborates the role of Urm1 adenylation for the complex formation. Mutants lacking ATP hydrolysis activity (R18A+R77A and D134A+Q164A) indeed show strongly decreased ATP-dependent Urm1 binding, whereas mutations of the active site cysteine that is known to form a thioester (C202S, C202A and C202K) are not affected. Therefore, we conclude that the adenylation of Urm1 is sufficient to stabilize the complex between Uba4 and Urm1 in the presence of ATP (Figure EV2D, Figure 3A and Appendix Figure S4).”*

In the previously submitted version, we have included a ATP-bound model of the Uba4-Urm1 complex (Fig. 3D upper panel). Considering the reasons provided above, we prefer to not speculate about the specific contributions of the ATP molecule itself and the individual phosphates to an increased affinity between Uba4 and Urm1.

3) In Table 1, *R*meas and *I*/sigmal values for the outer shell of CtUba4-CtUrm1 are too bad. Re-perform crystallographic refinement using lower resolution data (*I*/sigmal value for outer shell must be larger than 1.0). Provide *CC*_{1/2} value for outer shell in Table 1.

Response: We apologize for the accidental and unintended incompleteness of the provided crystallographic table (Table 1). We now re-evaluated our analyses and completed the table accordingly in the revised version of the manuscript. Foremost, we have successfully deposited both crystal structures with the PDB (6YUB and 6YUC) and provide fully annotated and approved pdb and mtz files together with the revised manuscript.

During crystallographic data processing, we generally follow the recommendations proposed by Karplus and Diederichs (Science. 2012; Acta Crystallogr D Biol Crystallogr. 2013 and Curr Opin Struct Biol. 2015). In detail, we used the percentage of correlation between intensities from random half-datasets (*CC*_{1/2}) value as a cut-off criterion, which must be significant at the 0.1% level (as reported by XDS log). We initially refined the CtUba4-CtUrm1 complex structure also including all collected data (3.08 Å; *CC*_{1/2}=0.167) and with different resolution cut-offs (e.g. 3.3 Å; *CC*_{1/2}=0.5) using the identical set of Rfree-flagged reflections. The obtained refinement statistics of *R*/*R*free 0.228/0.272 (3.08 Å) and 0.206/0.262

(3.3 Å) in combination with detailed inspection of the obtained density quality, initially guided us towards the optimal cut-off at 3.15 Å and R/Rfree 0.210/0.263. Following the recommendation of this reviewer, we also re-refined the structure using the cut-off criterium of $I/\sigma I = 1$ (3.66 Å). This resulted in slightly over-refined R/Rfree 0.18/0.25 values, obviously worse maps due to ignored information in the datasets and decreased model quality indicators (r.m.s.d. deviations in bond lengths and angles). In general, we are convinced that the proposed cut-off is approximately optimal and that too conservative cut-off criteria at this resolution would lead to the discarding relevant and useful data. Needless to mention, that hundreds of different crystals with worse diffraction properties have been tested before obtaining the analyzed dataset.

4) *In Table 1, B-factors for proteins in the complex are too large (189). Is there any reason for that? Provide a structural model colored with B-factors and non-biased omit map of the Urm1 C-terminus.*

Response: The B-factors for the proteins in the Uba4-Urm1 complex are relatively high, however the overall Wilson B factor value of the dataset is 158. Hence, the two critical values are in a similar range, which is also expected and common at this resolution. We followed the suggestion of reviewer 2 and re-analyzed our data with the “Staranis” and the “Anisoscalt” anisotropy servers. The analyses showed only mild anisotropy, but we nonetheless used the anisotropy corrected data for the structure refinement. By using reprocessed data from “Anisoscalt”, the B-factors of our refined model indeed dropped to 156 while the other refinement parameters virtually remained unchanged (see updated Table 1). Of note, we used map sharpening algorithms in Refmac to guide model building and we deposited both, anisotropy corrected data (PDB ID 6YUC) and original not modified SFs (PDB ID 6Z6S). We provide figures presenting the structure colored according to relative B-factor distribution (Figure EV3A) and Fo-Fc omit maps for the Urm1 C-terminus, the crossover loop and the peptide bond between these two components in the newly created Figure EV3B. We also updated the methods section to describe the anisotropy analyses and map sharpening.

Minor points

1) *In page 6, lines 14-16, "We found that ... This indicates that hydrolysis occurs between the alpha- and beta-phosphate", This logic is difficult to understand. Why could authors know the hydrolysis site based on the affinity difference? It may be possible that gamma-phosphate of ATP interacted with Urm1, which increased the affinity between Uba4-Urm1 (related to Major point 2).*

Response: Please see the elaborate answer and the results of additional experiments in response to point 2. We added a new paragraph on page 9 to clarify the description and interpretation of our data.

2) *In page 6, lines 18-19, "while AMP increased... indicating the formation of a stable adenylate intermediate", This sentence is confusing. E1 enzymes catalyze adenylation of Ubls using ATP and NOT AMP. It is impossible to form an adenylate intermediate of Urm1 using AMP.*

Response: We fully agree with this reviewer that the meaning of the original sentence was not completely clear. We think that the addition of AMP mimics the conformation of the nucleotide binding pocket, when adenylated Urm1 is bound and before the thioester intermediate is formed. In addition, we could also envision that during the transition from the adenylated form of Urm1 and the formation of a thioester intermediate with Cys202, AMP is actually formed and remains bound in the active site for a limited amount of time. Our findings that AMP itself strongly stabilizes Uba4 and the AD alone does not indicate the formation of a “stable intermediate” as written in the submitted manuscript, but indicates

that one of the AMP-related intermediates induces a more compact conformation of Uba4 resulting in a higher thermostability. We have now rephrased the respective section on page 6, which now reads as follows - *“AMP increased the thermal stability of full length Uba4 and the AD alone, indicating that an AMP-related intermediate, like the adenylated form of the Urm1 C-terminus induces a more compact conformation of the AD (Appendix Table S1).”*

3) In page 7, line 9, *“a strong hydrogen network”* should be *“a strong hydrogen-bond network”*.

Response: This has been corrected in the revised version of the manuscript.

4) In page 7, line 23, remove *“site”* after (C208A/C211A).

Response: The accidental misspelling has been corrected in the revised version of the manuscript.

5) In Table 1, provide units for B-factors.

Response: We have added the respective units for B-factors (\AA^2)

Referee #2:

The covalent modification of target proteins with ubiquitin or ubiquitin-like modifiers is initiated by ubiquitin/ubiquitin-like activating enzymes also referred to as E1 enzymes. These enzymes have evolved from simpler ancestors such as the E. coli MoeB and ThiF proteins, which catalyze the sulfur-incorporation steps during the biosynthesis of molybdopterin and thiamin, respectively. The fungal Uba4 enzyme and its human counterpart MOCS3 possibly represent a link between these activities since they catalyze a thiolation reaction, namely the conversion of uridin34 to thio-uridine in eukaryotic RNAs, and also the activation of a cognate ubiquitin-like protein, referred to as Urm1. The understanding of how these two activities are accomplished by one enzyme has been far from complete and this is where this manuscript makes an important contribution.

Pabis et al. describe crystal structures of Uba4 from Chaetomium thermoautotrophicum in its apo-state and in complex with Urm1. The structures are strengthened by biochemical structure-function studies in vitro but also in S. cerevisiae cells. Uba4 consists of an adenylation domain present in all E1 enzymes and also the MoeB/ThiF biosynthetic proteins and a unique rhodanese-like domain which, in its persulfide state, serves as sulfur donor for Urm1 thiocarboxylate formation. The apo-structure revealed an asymmetric arrangement, although both domains independently dimerize, however, the C2 axes are not aligned. In the Uba4-Urm1 complex the rhodanese-like domains become disordered and a C2 symmetric heterotetramer is formed. The linker region connecting the two domains of Uba4 was found to play a critical role during the catalytic mechanism.

Major concerns:

1. Crystallography:

a. The PDB validation reports were incomplete and lacked an independent confirmation of the reported R-factors. Maybe because no experimental data were submitted? If this assumption is correct, why not? Without this information, i.e. an independent confirmation of the reported R-factors I cannot endorse publication of this study.

Response: We did submit the experimental data together with the refined models to the PDB pre-validation server. Similar to this reviewer, we were surprised that the crystallographic R-factors are not re-calculated, but these are apparently only transferred from the information that is provided by the depositor. As the submitted pdb files were still before PDB deposition, they did not contain this information in their header section. We are very sorry for this confusion, but the provided R/Rfree values are in fact calculated from the provided experimental data and represent independent validations. Foremost, we have now deposited both structures to the PDB and we attach the full validation reports that we have received after the acceptance of our structures. The entries (6YUB and 6YUC) are currently on HOLD FOR PUBLICATION (HPUB) and have passed all stringent deposition criteria. In addition, we provide the annotated model (pdb) and experimental data (mtz) for the reviewers with the resubmission of the revised manuscript.

2. Table 2 is missing some key statistics, in particular values for CC(1/2) and R(pim). Without these statistical parameters I suspect that the resolution limit of the complex is too optimistic based on the values for R(meas) and CC(1/2) and R(pim) will allow the reader to conclude whether the high resolution cutoff makes sense. Since the authors did not mention whether either or both of these crystals diffracted X-rays anisotropically, one would assume that this is not the case. However, if there was significant anisotropy for one or both of the crystal forms, the data need to be reprocessed with the Staraniso server

from Global Phasing. Were NCS restraints (obviously domain-wise) employed during the refinement of the apo-structure? Even at 2.2 Å this will most likely improve model quality and possibly also narrow the gap between R and R(free) which for my taste is at the high end of what I would like to see at this resolution.

Response: We apologize for the accidental and unintended incompleteness of the provided crystallographic table (Table 1). We re-validated our analyses and completed the table accordingly in the revised version of the manuscript – now also listing R(meas) and CC(1/2) and R(pim). We used the “Staraniso” and the “Anisotropy” servers to analyze the Uba4-Urm1 dataset and found only mild anisotropy. Nonetheless, we used the anisotropy corrected data for the final structure refinement. As a result, the B-factors of our refined model of the complex decreased slightly to 156 while the other refinement parameters remained virtually unchanged (see updated Table 1). We assume that a R/Rfree gap of 4.57% for the Uba4 structure at 2.2 Å is reasonable and therefore assume that the usage of domain based NCS restraints is ultimately not necessary. Please also see the detailed response for similar issues raised by reviewer 1 and all relevant information has been included or updated in Table 1.

2. RHD-dimerization

a. Based on the available figures the interface between the AD dimer and RHD dimer seems to be minimal. This raises the question whether the arrangement and, in particular, the asymmetry seen here is a crystallization artifact? What are the packing interactions that the RHD dimer is involved in? How extensive are they, do they involve a symmetry-related AD dimer or RHD dimer or both? If they involve the AD dimer, could these interactions be of relevance also for an interaction in cis?

Response: A detailed inspection, did not display any direct influence from neighboring molecules in the RHD dimer interface. Furthermore, we have used the Evolutionary Protein-Protein Interface Classifier (EPICC; Bliven et al., PLoS Computational Biology 2018) to analyze all interfaces and molecular contact points in the found crystal lattices. These analyses included a comprehensive conservation analyses of the residues that participate in contact points, which clearly confirmed the biological relevance of the AD dimer and the interaction between Uba4 and Urm1. All other contact point lacked any significant sign of evolutionary selection pressure to conserve interface residues. Of note, the RHD domain of Chain A (containing the structured linker) does show a crystal contact (647 Å²) that might support its position. As the involved surface areas are not conserved, we believe that this interface has no biological relevance. Importantly the second RHD domain shows no crystal contacts with neighboring molecules and is held in place only via its interface with the other RHD domain. We have experimentally confirmed this interface in solution by size exclusion chromatography of the RHD alone and using interface mutants. The lack of packing restraints further corroborates the dimeric nature of the RHD in CtUba4. It is important to notice that the linker regions also show an asymmetric behavior that goes beyond the positioning of the RHDs. Overall, our structural data justifies the claim that Uba4 is asymmetric, even if crystal packing would lock a specific conformation.

b. The authors conclude (page 10, lines 7-9) without experimental evidence that the RHD dimer dissociates and in the discussion treat this as an observation and not as a hypothesis. Given the fact that the isolated RHD seems to stably dimerize considerable forces would be required to pull the RHDs apart. Where should that energy come from? It can only come from the binding of Urm1 and the energy would then not only be required for the dissociation of the RHD dimer but also for the displacement of the linker between the AD and RHD. Under normal circumstances I would request experimental verification of this

hypothesis, however, as I cannot judge whether it is possible for the authors to carry out experiments at all under the current circumstances, an alternative approach would be to state this as a hypothesis throughout the manuscript and point out that there is no experimental proof. In this context, it would be informative if the authors included the buried surface area upon RHD dimerization.

Response: We agree with the reviewer that our model is hypothetical at this point and that experimental validations would go beyond the scope of the revision. We have included question marks in the model of the reaction scheme (Figure 5A) and updated the text of the revised manuscript, which now reads as follows on page 10/11 – *“The symmetric nature of the Uba4-Urm1 complex and the mobility of the linker and the RHD form the basis of a hypothetical model where the RHD dimer that is observed in the Uba4 apo structure is disrupted upon Urm1 binding (Figure 3C). Subsequently, the individual monomeric RHDs may move towards the closest Urm1-bound AD. Furthermore, Urm1-induced monomerization of the RHD could simultaneously triggers exposure of the active-site cysteine (Cys397), which is buried in the RHD dimer interface and inaccessible in the Uba4 apo structure.”*

And on page 14 – *“...Our data allows us to draw a mechanistic reaction scheme of the individual steps leading to Urm1 thiocarboxylation by Uba4 and to propose the structural rearrangements in the AD and RHD domains upon substrate binding and product release (Figure 5A).”...“The binding of the Urm1 substrate to the AD enforces the displacement of the bound linker region, which could lead to the separation of the RHD dimer and structural rearrangements of the RHD domains.”*

In addition, we calculated the buried surface area using the PDBePISA server and now provide this in the text on page 5, which reads as follows *“The Uba4 RHD dimer shows a relatively small interface of 706.8 Å² and a unique arrangement compared to other known RHD dimers within the context of the full Uba4 dimer (Figure EV1C).”*

3. Binding order for ATP and Urm1

Based on thermal shift analyses in which the binding of different nucleotides to Uba4 was investigated the authors ultimately come to the conclusion (page 6, lines 23-25) that “This is in striking contrast to all other known canonical and non-canonical E1 enzymes that bind their UBL in the absence of ATP and Mg(2+)-ions”. Old kinetic data (Haas & Rose, JBC, 1982) postulate strictly ordered binding for Uba1 isolated from rabbit reticulocytes with ATP as leading substrate. This is certainly pertinent information in this context and should be mentioned/discussed. The previous sentence (lines 21-23) is trivial and is, of course, also true for every E1 enzyme, so please omit or rephrase. I also object to the statement that Urm1 does not bind to “ATP alone” (line 20) as the authors' own data contradict this conclusion. What the authors observe is a decrease in stabilization as one goes from AMP over ADP to ATP. From the manuscript I am also unclear whether Mg was present in these thermal shift assays, since there is no mention of Mg in the corresponding Materials and Methods section I would conclude that this is not the case, however, under these circumstances the data would be useless and have to be repeated with Mg being present. I am also confused regarding the statement (line 19) that the increased stability of the AMP complex indicates the formation of a “stable adenylate intermediate”. The acyladenylate intermediate can only form in the presence of the Urm1 C-terminus.

Response: All Uba4 proteins and variants were measured in the presence of 2mM MgCl₂ as described in the respective Materials and Methods section (“Thermal shift assay”). Our statement of uniqueness did not relate to the order of events and we fully agree that ATP has to be bound first similarly to the other representatives of E1 enzymes in the Uba family. We still did not observe binding of Urm1 to Uba4 in the

absence of ATP and presence of Mg²⁺ in two independent assays – GST-pull down (Figure 2A) and co-migration assay using size exclusion chromatography (revised Appendix Figure 4). We have rephrased the section on page 6/7 to adequately present our data and compare it to available literature, which now reads as follows – *“Thus, initial ATP binding (Haas & Rose, 1982) is followed by Urm1 binding, ATP hydrolysis and subsequent adenylation. Stable Uba4-Urm1 complex formation require the combined presence of Uba4, Urm1 and ATP in the reaction, which is in contrast to other known canonical and non-canonical E1 enzymes that are able to bind their UBL also in the absence of ATP and Mg²⁺ ions (Noda et al, 2011; Lake et al, 2001; Oweis et al, 2016).”*

4. Residue numbers and identities in Ct Urm1 vs. Sc Urm1

While Table S1 finally guides the reader in translating from the Ct residue numbers and identities to those of Sc, this is too late and too cumbersome for the reader. The authors need to come up with a better way to guide the reader through Figs. 2D, 4C and S8E as well as the corresponding sections in the text without having to look at this table over and over again.

Response: The reviewer touches an issue that has been intensively discussed among all authors before initial submission. Our study describes a large number of equivalent mutations in two different organisms and we are aware that differences in numbering can be confusing to the reader. The situation is further complicated by existing literature that exclusively uses the *S. cerevisiae* numbering, while our work mainly describes structural and biochemical work using the Uba4 and Urm1 proteins from *Chaetomium thermophilum*. Considering the comment and to improve clarity, we have decided to strictly stick to the nomenclature and residue numbering of CtUba4 and CtUrm1 in the text of this study. For all *in vivo* experiments describing *S. cerevisiae* versions of the proteins, we additionally indicate the yeast numbering in the figures using subscript (Figure 2, Figure 4, Figure EV5 and Appendix Figure S3).

Minor points:

1. Page 3, line 20: Replace "multi-domain" with "two-domain".

Response: The text has been rephrased accordingly.

2. Page 4, lines 15/16: Are there really structures available for all other eukaryotic E1 enzymes? I am pretty sure this is not the case.

Response: We did not want to indicate that structures are available for all other eukaryotic E1 enzymes, but state that there is at least one representative structure for every known eukaryotic E1 enzyme. We have rechecked the list of available structures in the PDB and found examples for Uba1, Uba2, Uba3, Uba5, Atg7, Nae1 and Sae1. We indeed did not find examples for Ubel2 (Uba6) and Ube1l (Uba7) and rephrased our statement on page 4 accordingly, which now reads as follows – *“Importantly, the molecular mechanism of Urm1 activation by Uba4 and the identity of its E2 and E3 enzymes have remained elusive, while structures are available for most other eukaryotic E1 enzymes.”*

3. Page 5, lines 6-8: Please rephrase, e.g. "While the AD and RHDs form C2-symmetrical dimers the structure is asymmetric since the two 2-fold axes do not coincide." or something similar.

Response: We believe that our initial sentence is addressed to the broad readership, but we now included a second sentence similar to the suggested one on page 5 of the revised manuscript that reads as follows – *“While the ADs and RHDs individually form C2-symmetrical dimers the structure is asymmetric since the two 2-fold axes do not coincide.”*

4. Page 5, line 19 and whenever this is present in one of the figures: What is an "active loop"? Since the Cys is part of the loop it probably should be called "active site loop".

Response: The accidental misspelling has been corrected in the revised version of the manuscript.

5. Page 8, line 15: Please state that the nine residues in the linker region were replaced with three Gly-Gly-Ser repeats, at least that was my interpretation.

Response: The interpretation is correct and the text has been rephrased accordingly.

6. Page 10, lines 3-4: It might be helpful to state that the twofold axis of the Uba4-Urm1 complex is a crystallographic symmetry element.

Response: In this context, we would like to use the word "symmetric" for the description of the Uba4-Urm1 dimer. To avoid repetition of words, we decided to show the symmetry operator in Figure 3B and Figure 3C and add a description in the respective figure legend.

7. Page 10, line 7: Replace "from one molecule" with "from the one molecule where it is bound"

Response: The sentence has been rephrased accordingly.

8. Page 10, lines 7-9: I fail to see how the second sentence is a logical conclusion from what is stated in the first sentence.

Response: The sentence has been rephrased accordingly.

9. Page 11, line 25: Are these two cysteines visualized in one of the figures? If yes, reference here, if not, include a suitable figures or modify an existing figure accordingly.

Response: Neither Cys202 nor Cys305 are directly visible in the structure of apo-Uba4. In the Uba4-Urm1 complex structure position 202 gets structured, but as we used a C202K mutant we can "only" see the position of Lys202 (and not Cys202) covalently linked to the C-terminus of Urm1. As also described in the reply to point 25 of the reviewer "Cys305 is disordered in both structures, but its approximate position in the apo Uba4 structure close to the active site can be deduced". We now labeled the approximate position of Cys202 and Cys305 in the lower panel of Figure 3D to illustrate the tentative positions of Cys202 and Cys305 in the respective structures. We believe their proximity and position in flexible loop regions would permit a direct communication between these two residues. We now added the figure reference on page 12 of the revised version, which now reads as follows – "Surprisingly, Urm1 conjugation is reduced to wild-type levels in the Uba4C202S/C305S double mutant (Figure 4B), indicating a crucial redox communication between these two cysteines both located in adjacent loop regions (Figure 3D)."

10. Page 12, line 2: "Cys311 is less critical ..." Less critical than what exactly?

Response: The text has been rephrased to make this point more clear.

11. Page 14, lines 22-24: This part of the discussion would suggest that the reaction catalyzed by Uba4 and the evolutionary ancestors of the E1 family, MoeB and ThiF, are more complex than let's say the E1 for ubiquitin and that these modern E1 have evolved in such a way as to prevent interactions with rhodanases.

Response: We agree with the interpretation of the reviewer, but would not state that Uba4 and the mentioned ancestors are more complex. The dual function of Uba4 as a SCP and UBL might give this impression. However, the modern E1 systems may be able to carry out more specialized functions. After all, the Uba4-Urm1 system lacks E2 and E3 enzymes, which add several additional layers of complexity to the eukaryotic UBL systems.

12. Legend to Fig. 1

(A) Rephrase to "rhodanese-like domains (RHDs) of either Tum1 or Uba4." to reflect the arrows drawn in the figure.

(B) Add "RHD: Rhodanese-like domain"

(D) "Dotted line" should be "dashed line" and "(middle)" should be replaced with "(central inset)" or something similar.

Response: The figure legend has been changed accordingly.

13. Fig. 2B: Highlight Asn307 like the other residues which have been studied

Response: We have changed the labeling of the residue accordingly.

14. Legend to Fig. 2(B) Add "(for reference see full structure on upper left)" after "Close up" and after "zinc site" add "(inset, lower left)".

Response: The figure legend has been rephrased accordingly.

15. Legend to Fig. 3: (C) Insert "which form the dimer" after "adjacent asymmetric unit"

Response: The figure legend has been rephrased accordingly.

16. Fig. 4D: Will this be readable, even if shown at higher resolution?

Response: The panel of figure has been relabeled and font size adjusted.

17. Legend to Fig. 5: (B) Add "unless they are covalently linked" after "are indicated."

Response: The figure legend has been rephrased.

18. Table 1

- " $|/\text{sig}|$ " must be ""
- I doubt that a precision to 1/1000 Å is warranted for the unit cell dimensions of the apo-structure.
- Which ligands/ions are present in the apo-structure? With an average B-factor of >187 Å

Response: We have re-inspected the structure and have found that a single molecule of ethylene glycol (present in the cryo-protection solution) has been placed, but is most likely not sufficiently supported by robust electron density. Hence, we decided to remove this ligand molecule and to re-refine the structure. The B-factor value for the remaining ligands is 41.72 Å².

(2) the question arises, are these ligands really present?

Response: After removal of ethylene glycol molecule, the only ligands present in the structure are two Zinc ions, which are well coordinated and show lower B-factors than the overall structure. The atom number and B-factor values for ligands has been updated in Table1.

19. Page 29, line 23: Briefly describe why the C55S variant was used.

We removed a non-conserved surface cysteine in Urm1 from *Chaetomium thermophilum* to avoid artefacts during the detection of the thiocarboxylated C-terminus by polyacrylamide gels containing [(N-Acryloylamino)phenyl]mercuric Chloride (APM). We added a short note in the respective materials and methods section on page 18, which reads as follows – “We removed a non-conserved surface cysteine in CtUrm1 to circumvent artefacts during the detection of the thiocarboxylated C-terminus using [(N-Acryloylamino)phenyl]mercuric Chloride (APM) containing SDS-PAGE.”

20. Page 30, line 18: Insert "published protocols" after "according to".

Response: The text has been rephrased accordingly.

21. Page 32, line 8: Since "maximal size" is mentioned, how big were the crystals?

Response: We added the approximate crystal dimensions on page 20.

22. Page 32, line 14: How was Urm1 modeled? By MR or built de novo or positioned as a rigid body in the map?

Response: CtUrm1 was placed by MR based on the known ScUrm1 structure, the sequence manually curated and then refined.

23. Fig S3A: Use decimal points instead of commas.

Response: The accidental use of commas has been corrected in the revised version of the figure (now Expanded view figure 1).

24. Legend to Fig. S3 (C) Please define "BtTST".

Response: The description has been added accordingly (now Expanded view figure 1).

25. Fig.S4B: Label first and last residue of loop as well as residues, which are adjacent to the gap. Is the dashed link representing the disordered residues shifted to the right? It certainly does not connect the residues adjacent to the gap.

Response: The dashed line indicating flexible residues has indeed been accidentally moved during the assembly of the figure. We now corrected the position of the loop and indicated the last visible residues. After labeling it also becomes obvious that Cys305 is disordered in both structures, but its approximate position in the apo Uba4 structure close to the active site can be deduced.

26. Fig. S6B: Add gel analysis for the fractions from the Uba4-Urm1 SEC run in the absence of ATP.

Response: We have repeated the analyses and now show comparative gel filtration profiles for Urm1+ATP, Uba4+ATP and Uba4-Urm1 +/- ATP. Furthermore, we show SDS-PAGE analyses of the fractions from the mixed Uba4-Urm1 samples in the presence and absence of ATP. The analyses clearly show that the addition of ATP is required for a stable interaction between Uba4 and Urm1. The new data

replaced the previous figure panels A and B in Appendix Figure 4. We have updated the respective figure legend accordingly.

27. Fig. S7

(A) Right panel: Instead of showing a crowded superposition containing two maps show them side-by-side.

Response: The previously presented “maps” were actually the same refined 2Fo-Fc map, colored differently in the distinct separate regions of the protein complex. We recognized that this representation was potentially confusing and misleading. In accordance with this comment and the question 4 of reviewer 1, we have calculated omit Fo-Fc maps of the different areas and compared them to the refined maps of the same regions to facilitate the interpretation. A detailed presentation of these results can be found in the newly introduced Figure EV3B of the revised manuscript. We would like to highlight that pdb and structure factors will be publicly available for the interested reader to independently judge our map interpretations.

(B) Replace "Full-length Uba4" with "Apo-Uba4".

Response: We have replaced "Full-length Uba4" with "Apo-Uba4" in Figure 3B as well as in the newly created Figures EV3A and EV4A.

Referee #3:

The manuscript " Molecular basis for the 2 bifunctional Uba4-Urm1 sulfur relay system in tRNA thiolation and ubiquitin-like conjugation" by Pabis. et al. reports the crystal structures of the full length Uba4 alone and in complex with Urm1. These structures are complemented with biochemical analysis including structure-based mutations that are tested for their effects on yeast growth. Based on their results the authors suggest that the apo Uba4 forms an asymmetric dimer, and both the AD and RHD contribute to this structure. In this structure, the RHD is inactivated. Moreover, using pull down experiments they show that Uba4 does not bind Urm1 if ATP is absent. Interestingly, the linker connecting the AD to the RHD is visible only in one molecule of Uba4. In that molecule, part of the linker is in close proximity with the ATP binding site. The authors suggest that the linker coordinates the Uba4 catalytic steps starting from ATP binding to thiocarboxylation. Finally, in the structure of Uba4 linked to Urm1 via an isopeptide bond, they show that Uba4 -Urm1 forms a symmetric dimer. In this structure the dimerization of the RHD is disrupted. This, in turn, exposes the active site Cys of RHD which is buried in the apo structure.

Taken together, this work advances our understanding of Uba4 activity and provide novel structural mechanism for the crosstalk between the AD and the RHD. However, the data supporting the conclusions require further validation.

Major points:

1. The authors performed pulldown experiments with GST-Urm1 and Uba4 to show that Uba4 binds Urm1 only in the presence of ATP. They also argue that this is unique to Uba4 compared to other E1 enzymes (page 6; 23-25). Based on their results one cannot rule out the possibility that the increase of binding in the presence of ATP is due to the formation of thioester bond between Urm1 and Uba4 active site (Cys202). This stabilizes the interaction between the two proteins. Finally, when the complex is loaded on the gel and breaks down due to reducing agent, the intensity of the Uba4 increases and this leads to the conclusion that ATP increases the affinity between the two proteins. This is not unique to Uba4 and holds true with other E1 enzymes and their Ubls. Therefore, in order to rule out this possibility the authors have to redo the pulldown experiments with Uba4 C202A that cannot form thioester bond with Urm1.

Response: We fully agree with the interpretation of this reviewer and assume that the increased affinity between Urm1 and Uba4 is not caused by the presence of an ATP molecule itself. As outlined in the response to a similar concern by reviewer 1, we have now performed additional experiments using a C202A mutant of CtUba4. We can show that indeed different mutants that affect the ATP hydrolysis rate decrease the ATP-dependent complex formation (R18A+R77A and D134A+Q164A), whereas Cys202 mutants do not show this decrease in the presence of ATP (Figure EV2D). Hence, our data suggests that adenylation of the Urm1 C-terminus stabilizes the complex between Uba4 and Urm1. Although the formed thioester by Cys202 is crucial for the activity of Uba4 and the self-protection of Uba4 against Urm1-COSH induced self-conjugation, it does not seem to be necessary for the stabilization of the complex. Furthermore, we have now added dedicated co-migration assays (Appendix Figure S4A) that clearly show the ATP-dependency of the complex formation. We have rephrased the statement of uniqueness to the comparison with the E1-UBL systems that are known to bind each other even in the absence of ATP or functional adenylation. The sentence on page 6/7 now reads as follows – *“Stable Uba4-Urm1 complex formation require the combined presence of Uba4, Urm1 and ATP in the reaction, which is in contrast to other known canonical and non-canonical E1 enzymes that are able to bind their*

UBL also in the absence of ATP and Mg²⁺ ions (Noda et al, 2011; Lake et al, 2001; Oweis et al, 2016)." (please see also response to comment 2 of reviewer 1 and comment 3 of reviewer 2)

2. *The authors suggest the linker contacting the AD to the RHD is critical for the proper functioning of Uba4. Based on the structural data they suggest that it affects ATP binding. These conclusions are based on the structure of the linker in one Uba4 molecule but not in the other. My question is how the authors know that the structured linker in one of the molecules is not due to crystal packing? One possibility is that the mutations in the linker are structural mutations that affect the protein structure, similarly to the mutation in the Zn binding site.*

Response: We have analyzed the crystal packing of the apo-Uba4 structure and did not identify any direct contacts of the linker region with neighboring molecules. In the initially submitted version of the manuscript, we have provided thermostability data for all tested mutants in the absence of Urm1 and nucleotides, including the mutants located in the linker region and the AD alone. In detail, we did not detect clear destabilizing effects of any linker mutations (now found in Appendix Table S1). Furthermore, we tried to support our claim more directly by comparing the stability and ATP hydrolysis activity of the AD domain alone with a newly generated construct that also contains the linker region (AD_{linker 1-321}) and only lacks the RHD domain. We also engineered a construct that adds only those linker residues that are bound in the cavity of Uba4 (AD_{linker 1-315}) and compared all constructs to full length Uba4. Our previously presented data already showed that the AD domains shows a strongly reduced stability and ATP hydrolysis activity compared to the full length Uba4 protein. The newly included data clearly shows that the addition of the linker region is sufficient to stabilize the Uba4 protein in the absence of Urm1 and nucleotides (Appendix Table S1) and to restore the same ATP hydrolysis activity in the presence of Urm1 as the full length protein (Figure 2 and Appendix Table S2).

We updated the figures and the section on page 8 of the manuscript, which now reads as follows - *"Furthermore, we attached the linker residues that are expected to compete with Urm1 binding (AD_{linker 1-315}) and the complete linker region (AD_{linker 1-321}) to the AD and compared their stability and their activity to full length Uba4 and the AD alone. We show that the linker region strongly enhances the thermostability of the AD domain in the absence of Urm1 and that it stimulates the ATP hydrolysis activity of the AD in the presence of Urm1 (Figure 2C and Appendix Table S1)."*

3. *It is not clear why the authors decided not to test the same mutations both in the in vitro and in vivo assays. For example, the D134A mutation which has strong effect on ATP hydrolysis was not checked in thiolated tRNA or yeast growth. This is only one example. Please add this information.*

Response: The equivalent mutation of D134A in yeast (D166A) has been tested and the results have been shown in Figure 2D. In detail, CtUba4 D134A shows a strongly reduced ATP hydrolysis activity *in vitro* and the equivalent ScUba4 D166A mutant in yeast shows almost completely abolished tRNA thiolation levels. In general, there is a very high correlation between the effects observed *in vitro* and *in vivo*. We believe that this question partially relates to the numbering and labeling issues (also raised by reviewer 1 question 4). We hope that the newly introduced labeling scheme reduces the confusion between equivalent mutations in *S. cerevisiae* and *Chaetomium thermophilum*. Of note, due to experimental complexity of yeast strain generation, we focused on the most relevant strains and selected mutants in accordance to the results obtained *in vitro*.

4. *The linker is critical for the catalytic activity of Uba4 (page 8; 17-18). If this is the case, please show that the AD domain alone does not hydrolyze ATP.*

Response: In the previously submitted version, we have included data on the activity of the AD domain alone (Table S2), showing that the AD domain shows strongly decrease ATP hydrolysis activity. As described above, we generated additional constructs to validate our hypotheses experimentally. We now show that parts of the linker are sufficient to stabilize the AD domain and restore the ATP hydrolysis activity – directly showing that the linker has an influence on stability and activity of Uba4 in the presence and absence of Urm1. We added a short section describing these data on page 8 of the manuscript (see response to comment 2 above).

5. The mutations N307/M308 to Ala stimulates ATP hydrolysis. Is it because the protein is more stable? and not because it mimics URM1 binding. Also, these residues are not conserved in yeast (table S1)

Response: In the previously submitted version, we have included data on the thermostability of all tested mutants, including CtUba4 N307/M308. The mutant shows an almost identical stability profile as the wild type Uba4 protein. Other variants with an identical stability profile (e.g. D144A) did not stimulate ATP hydrolysis.

6. It is not clear why the mutation of C397S but not C202 affects ATP hydrolysis. Following the model (Fig 5), C397 functions after the formation of the thioester bond with C202, so how can C397 affect hydrolysis but not C397?

Response: This observation was indeed surprising and the detailed underlying mechanisms remain to be determined. It should be highlighted that other mutations in the RHD active site (R398A+R399A and Q201A+D402A) also stimulated ATP hydrolysis, supporting the functional connection independent of Cys397. As mentioned in the discussion on page 14, we consider the possibility that “...Urm1 remains in the same location during the final thiocarboxylation reaction as well, since Urm1 binding triggers the release of the flexible linker region and hence the movement of the RHDs. Therefore, the active site of the RHD rather moves into proximity of the C-terminus, than Urm1 shuttling to the RHD domain (Figure 5A).” The structural rearrangements in this model would allow bringing the sites of ATP hydrolysis in the AD and the active site of the RHD (Cys397) into close proximity (see model in Figure 5). It will require additional experimental proof to establish this model, but this is beyond the scope of this manuscript. We have rephrased and emphasized the hypothetical nature of this model in response to next comment and comment 2 of reviewer 2.

7. The authors suggest that in the Uba4-Urm1, the RHD is no longer a dimer. Moreover, although the RHD is not observed in the structure they claim that C397 of the RHD is getting next to C202. This is too speculative and has to be shown.

Response: We have rephrased all sections on pages 11 and 14 to emphasize the theoretical nature of the proposed model. In addition, we have added question marks in the model of the reaction cycle (Figure 5A) to tone down these claims. We are very eager to answer this question and get further structural insights into the mechanistic understanding of the ongoing domain rearrangements. Nevertheless, we hope that the reviewer understands that providing these answers would go beyond the scope of this revision and manuscript (also see response to comment 2b from reviewer 2).

8. In Fig 4a why ATP is present only with the WT Uba4 and not with the C202S mutation? Does Urm1 form oxyester bond with Uba4 C202S. What is the effect of the C202A mutation? Does it behave as C202S?

Response: We do not need to add ATP to form the observed covalent lysine conjugates formed by Urm1-COSH in combination with Uba4 C202S under mild oxidative stress conditions. By adding the product (Urm1-COSH) to Uba4, we mimic the product release state rather than proceeding through the whole reaction cascade. We included the controls for Uba4 WT to show that Urm1-COSH does not form aggregates, if Cys202 is present and that Urm1-OH only covalently attaches to Uba4 C202K after successful adenylation after addition of ATP. Following the questions by this reviewer and reviewer 1, we have purified the requested C202A mutant and show that it behaves identically to the C202S mutant in all performed assays. The new data for Uba4 C202A is now included in Figure 4G and Figure EV2D (see response to comment 1 of reviewer 1)

Minor points

1. *Figure 1A - it should be AD and not UBA4*

Response: The labeling in the figure has been corrected accordingly.

2. *Figure 1B - why the active site Cys397 of RHD is not shown*

Response: The labeling in the figure has been changed accordingly.

3. *What is the surface representation in fig 1D? why not to have the AD and AD' at a different color?*

Response: The AD dimer is emphasized by a transparent surface representation and we would prefer to keep the current representation and color scheme. The description of the figure legend now includes a comment on the surface representation.

4. *Please provide the rmsd of the superpositions in Fig 1E*

Response: The RMSD values have been added to the figure and the figure legend has been amended accordingly.

5. *Fig2B the label for Asn307 is missing*

Response: The labeling in the figure has been corrected accordingly.

6. *Fig3C please add in the legend the red and green colors.*

Response: The figure legend has been amended accordingly.

7. *Please add cc1/2 value for the crystallographic table*

Response: The table has been completed accordingly.

Dr. Sebastian Glatt
Jagiellonian University
Malopolska Centre of Biotechnology
Gronostajowa 7a str
Krakow 30-387
Poland

17th Jun 2020

Re: EMBOJ-2020-105087R

Molecular basis for the bifunctional Uba4-Urm1 sulfur relay system in tRNA thiolation and ubiquitin-like conjugation

Thank you for submitting your revised manuscript to The EMBO Journal. All three original reviewers have now looked at it again, and found their key concerns generally satisfactorily addressed. They still retain a few minor/specific issues (see below), which I would ask you to respond to and/or incorporate during a final round of revision; additional experiments (such as proposed by referee 3) will not be required, but please consider whether the suggestion for structural data re-analysis made by referee 1 would be possible.

During this final modification, please also address the following editorial points:

REFEREE REPORTS

Referee #1:

The authors have addressed almost all of my concerns. One remaining concern is about the refinement of the Uba4-Urm1 complex. The authors state that Rfree values become 0.272, 0.263, 0.262 and 0.25 for refinement at 3.08, 3.15, 3.3, and 3.66 Å resolution, respectively, which is not consistent with the general tendency that Rfree values become lower when higher resolution data are used for refinement. The best Rfree value at 3.66 Å resolution (0.25) would mean that the quality of the data at the outer shell is seriously bad due to the too weak signals (I/σ is below 0.5). Overfitting at 3.66 Å resolution ($R=0.18$ vs $R_{\text{free}}=0.25$) could be overcome by setting the crystallographic refinement parameters optimal (Phenix can resolve the problem of overfitting than Refmac). Therefore, this reviewer proposes that the authors re-perform refinement using data in resolution range that provides the best Rfree value without overfitting.

Referee #2:

The authors have substantially revised the manuscript taking into account my criticisms raised in the initial round of review and, as far as I can tell, also the concerns listed by the other reviewers. Going through the manuscript I only spotted the three minor issues listed below.

Minor points:

- (1) The overall $R(\text{pim})$ values given in Table 1 appear to be incorrect, i.e. 0.54% should be 5.4% and 0.21% should be 2.1%?
- (2) The almond-shaped symbol in Fig. 3C is incorrect (it actually indicated a twofold axis perpendicular to the plane of the paper) and should be replaced with a vertical double-headed arrow (which designates a twofold axis in the plane of the paper).
- (3) I believe the past tense of the verb "to lead" is "led", which has not been used in a couple of instances.

Referee #3:

The revised manuscript is improved with further data including additional Uba4 mutations. The authors show that Uba4 C202A does not prevent ATP from increasing the affinity of Uba4 to Urm1. However, to show that the effect of ATP is due to the formation of Urm1~AMP, and not just ATP binding, the authors can use the Urm1 mutant G111C, which, doesn't trigger ATP hydrolysis, and thus no Urm1-AMP formation.

The data supporting the movement of RHD upon Urm1 binding are still not very solid. However, in the revised manuscript the authors tone down their claims and clearly point out in the text that their model of RHD movement is hypothetical. Also, in the figure they included question marks to highlight the missing data.

Point-by-Point Response

EMBOJ-2020-105087R

Referee #1:

The authors have addressed almost all of my concerns. One remaining concern is about the refinement of the Uba4-Urm1 complex. The authors state that Rfree values become 0.272, 0.263, 0.262 and 0.25 for refinement at 3.08, 3.15, 3.3, and 3.66 Å resolution, respectively, which is not consistent with the general tendency that Rfree values become lower when higher resolution data are used for refinement. The best Rfree value at 3.66 Å resolution (0.25) would mean that the quality of the data at the outer shell is seriously bad due to the too weak signals (I/sigma is below 0.5). Overfitting at 3.66 Å resolution (R=0.18 vs Rfree=0.25) could be overcome by setting the crystallographic refinement parameters optimal (Phenix can resolve the problem of overfitting than Refmac). Therefore, this reviewer proposes that the authors re-perform refinement using data in resolution range that provides the best Rfree value without overfitting.

Response: We have worked with this specific dataset for almost 2 years and we have applied many more processing routes than those mentioned in the manuscript or suggested by the reviewers – including Phenix and Buster. Regarding the final comment of the reviewer - “Therefore, this reviewer proposes that the authors re-perform refinement using data in resolution range that provides the best Rfree value without overfitting.” This is exactly what we have done previously, following the reviewer’s recommendation and described in the first point-by-point response (comment 3 of reviewer 1). We are still convinced that the resolution cutoff at 3.15 Å is well justified and that we would ignore useful information that can aid model building for structures at low resolution. Nonetheless, we re-refined the final model with phenix.refine 1.18.2 and also used automated weight optimization, like for almost all of our other crystallographic studies. Of note, we obtained much higher R/Rfree values at 3.15 Å (0.249/0.295) and 3.66 Å (0.232/0.284) resolution (as previously observed). Furthermore, automated weight optimization in phenix did slightly improve stereochemical parameters, but did not improve the R/Rfree values at 3.15 Å (0.251/0.298) and 3.66 Å (0.214/0.263) and caused an increase of Ramachandran outliers. Although, anisotropy corrected data was used (as previously suggested by the reviewer), the B-factor values exceed 190 Å² in all phenix runs – compared to the previously optimized values of 156.74 Å². It is not entirely clear to us what the software-specific reasons for the detected differences are. However, we assume that Refmac, which also has automated map sharpening tools incorporated, deals better with this dataset. We would also like to highlight that the resulting atomic models and refined densities from Refmac, Phenix and Buster are indeed very similar.

We know that even more processing steps and approaches could be used, but neither the structure, the map quality nor the resulting structural conclusions will change substantially (like in the case of the previously performed anisotropic B-factor refinement). Without the indications of a clear added scientific value and the wealth of complementary biochemical validation experiments, we hope that the reviewer agrees with publication of our manuscript in its current form.

Last but not least, we would like to highlight that pdb and original structure factors will be publicly available for the interested reader to independently judge our map interpretations using different analysis tools.

Referee #2:

The authors have substantially revised the manuscript taking into account my criticisms raised in the initial round of review and, as far as I can tell, also the concerns listed by the other reviewers. Going through the manuscript I only spotted the three minor issues listed below.

Minor points:

(1) The overall R(pim) values given in Table 1 appear to be incorrect, i.e. 0.54% should be 5.4% and 0.21% should be 2.1%?

Response: The reviewer is correct, and the accidental misspelling has been corrected in the revised version of the crystallographic table.

(2) The almond-shaped symbol in Fig. 3C is incorrect (it actually indicated a twofold axis perpendicular to the plane of the paper) and should be replaced with a vertical double-headed arrow (which designates a twofold axis in the plane of the paper).

Response: The symbol has been changed to a vertical double-headed arrow, representing a twofold axis in the plane of the paper.

(3) I believe the past tense of the verb "to lead" is "led", which has not been used in a couple of instances.

Response: The wording has been corrected from "lead" to "led" on page 6 (line 16) and page 10 (line 8).

Referee #3:

The revised manuscript is improved with further data including additional Uba4 mutations. The authors show that Uba4 C202A does not prevent ATP from increasing the affinity of Uba4 to Urm1. However, to show that the effect of ATP is due to the formation of Urm1~AMP, and not just ATP binding, the authors can use the Urm1 mutant G111C, which, doesn't trigger ATP hydrolysis, and thus no Urm1-AMP formation.

The data supporting the movement of RHD upon Urm1 binding are still not very solid. However, in the revised manuscript the authors tone down their claims and clearly point out in the text that their model of RHD movement is hypothetical. Also, in the figure they included question marks to highlight the missing data.

Response: We do agree with the reviewer that the analyses of an Urm1_{G111C} (or an Urm1ΔG) mutant will be very interesting in this context. Nonetheless, we are still lacking a reliable assay to estimate the binding affinity for ATP alone (see previous response to comment 2 of reviewer 1). Therefore, results from these mutants would still be ambiguous. For instance, if these variants would not bind to Uba4 in the presence of ATP (as expected), we would still need to directly show that ATP binding is not affected. We agree that these additional experiments will in principle be interesting, but we believe that the establishment of these assay would be beyond the scope of this manuscript at this stage. We hope that the reviewer appreciates our previous efforts to discriminate between adenylation and thioester formation and will support publication of our manuscript in its current form without additional data from the suggested mutants – as also indicated by the editor.

Dr. Sebastian Glatt
Jagiellonian University
Malopolska Centre of Biotechnology
Gronostajowa 7a str
Krakow 30-387
Poland

26th Jun 2020

Re: EMBOJ-2020-105087R1

Molecular basis for the bifunctional Uba4-Urm1 sulfur relay system in tRNA thiolation and ubiquitin-like conjugation

Thank you for submitting your final revised manuscript for our consideration. I am pleased to inform you that we have now accepted it for publication in The EMBO Journal.

Corresponding Author Name: Sebastian Glatt

Journal Submitted to: The EMBO Journal

Manuscript Number: EMBOJ-2020-105087